# FACT: Fuzzy Alignment with Comorbidity Topology for Reliable Multi-Label Medical Image Diagnosis

**Yingyu Chen** [1]  **Yongqiang Huang** [2]  **Yang Qin** [1]  **Ziyuan Yang** [1]  **Lang Yuan** [2]  **Maosong Ran** [1]  **Yi Zhang** [2]

## Abstract

In clinical practice, patients often present with multiple co-occurring diseases, yet most existing Multi-Label-Diagnosis (MLD) methods treat diagnosis as a rigid discriminative partitioning task, implicitly assuming that overlapping pathologies are separable. This assumption is problematic in medical images, where identical or highly similar visual observations may simultaneously support multiple disease labels, and disease concepts are inherently correlated rather than independent. Enforcing hard decision boundaries under such overlap suppresses shared evidence, biases feature representations, and ultimately undermines model reliability. To address this limitation, we propose **F**uzzy **A**lignment with **C**omorbidity **T**opology (**FACT**), a novel paradigm that reformulates MLD as a fuzzy alignment problem between atomic visual evidence and disease semantic anchors. FACT is characterized by three key features: (1) modeling visual polysemy through shared and reusable atomic visual evidence; (2) encoding disease correlation via semantic anchors structured by comorbidity topology; and (3) employing a metric-based fuzzy membership function for non-discriminative visual-semantic alignment. Extensive experiments on three public clinical benchmarks, together with additional evaluation under long-tailed and noisy settings, demonstrate that FACT consistently improves diagnostic performance while delivering clinically plausible predictions. The code is available at https://github.com/yyuChen9/FACT.

---

[1]College of Computer Science, Sichuan University, Chengdu, China, 610065 [2]School of Cyber Science and Engineering, Sichuan University, Chengdu, China, 610207. Correspondence to: Yi Zhang <yzhang@scu.edu.cn>.

*Proceedings of the $43^{rd}$ International Conference on Machine Learning*, Seoul, South Korea. PMLR 306, 2026. Copyright 2026 by the author(s).

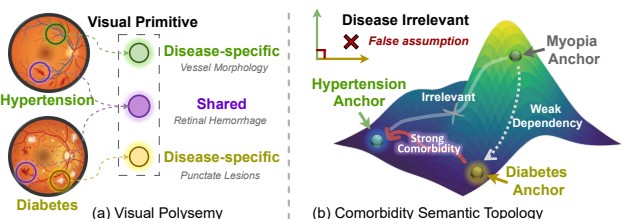

*Figure 1.* (a) Visual Polysemy. Pathological signs behave as compositional atoms rather than exclusive features. (b) Comorbidity Semantic Topology. Diseases lie on a structured semantic manifold shaped by physiological comorbidity relationships.

## 1. Introduction

Medical image diagnosis plays an indispensable role in clinical practice, aiding in the early detection and monitoring of complex pathologies. A single medical image often contains visual evidence corresponding to multiple co-occurring diseases. For example, chest radiographs may simultaneously exhibit signs of cardiomegaly, pulmonary edema, and pleural effusion, while fundus images can reflect overlapping pathological patterns associated with different retinal disorders. As a result, multi-label medical image diagnosis (MLD) has emerged as a fundamental and widely adopted task in medical image analysis (Litjens et al., 2017). While deep learning models have achieved expert-level performance on single-disease benchmarks (Esteva et al., 2017; Tiu et al., 2022), MLD remains fundamentally challenging, as comorbidity introduces complex visual and semantic dependencies among diseases, rather than isolated and independently observable conditions. Addressing these challenges is crucial for developing models that deliver both accurate and clinically reliable predictions in real-world settings (Yang et al., 2021).

Existing MLD methods mainly follow network-centric and objective-centric paradigms. Network-centric approaches (Zhang et al., 2023; Hou et al., 2025; Öztürk et al., 2025) typically employ hybrid CNN–Transformer architectures to extract global visual representations, aiming to enhance feature expressiveness. In parallel, objective-centric strategies (Park et al., 2023; Kobayashi, 2023) design inter-disease contrastive objectives to explicitly maximize disease-wise discriminability. More recently, several works (Wang et al.,

2022; Huang et al., 2024) exploit knowledge learned from large pre-trained models, such as ViT (Dosovitskiy, 2020) and CLIP (Radford et al., 2021), to introduce semantically grounded supervision for MLD. Despite these architectural and training innovations, most existing methods ultimately reduce MLD to global feature aggregation with independent binary relevance modeling. This formulation implicitly enforces rigid decision boundaries, assuming separable feature geometry and weak coupling among disease concepts. In practice, it is commonly instantiated via independent sigmoid classifiers optimized with binary cross-entropy or its variants, which explicitly encourage per-disease separability in the feature space.

We argue that this rigid alignment formulation contradicts two intrinsic properties of medical images: **visual polysemy** and **disease semantic correlation**. First, as illustrated in Figure 1(a), pathological primitives are inherently compositional, such that a single visual observation can serve as shared evidence supporting multiple diagnostic labels simultaneously. Enforcing exclusive decision boundaries under this property incentivizes models to suppress shared visual evidence in order to satisfy discriminative objectives, leading to biased representations. Second, disease concepts are not independent but exhibit structured and often asymmetric semantic correlations rather than orthogonal category partitions. As depicted in Figure 1(b), diabetes can lead to hypertension, whereas the presence of hypertension does not imply diabetes. Treating disease labels as isolated implicitly assumes orthogonal disease semantics, which contradicts clinical practice and limits model reliability.

Motivated by these observations, we reformulate MLD as a fuzzy alignment problem that models non-exclusive associations between visual evidence and disease concepts. To instantiate this idea, we propose **F**uzzy **A**lignment with **C**omorbidity **T**opology (**FACT**), a novel paradigm that explicitly constructs shared and reusable atomic visual evidence and disease semantic anchors encoded by comorbidity topology, and aligns them in a unified embedding space. On the visual side, FACT organizes visual observations into an Atomic Visual Space via vector quantization, yielding shared and reusable atomic visual evidence that captures visual polysemy. On the semantic side, FACT builds a Comorbidity Topology from the disease correlation matrix and refines disease name embeddings using a graph convolutional network, producing correlation-aware semantic anchors that reflect structured and asymmetric disease relationships. Based on these representations, FACT adopts a metric-based fuzzy membership function derived from variational principles, which enables each visual atom to provide partial, concurrent, and non-exclusive support to multiple disease categories. This formulation yields a unified and mathematically grounded framework for multi-label diagnosis that naturally accommodates visual overlap and disease

correlation.

Our main contributions are summarized as follows:

1. We reveal a fundamental mismatch between rigid discriminative learning and the intrinsic properties of medical images, which motivates a reformulation of MLD as a fuzzy alignment problem between atomic visual evidence and disease semantic anchors.

2. We propose FACT, a novel paradigm that explicitly models visual polysemy and disease semantic correlation by constructing shared atomic visual evidence and comorbidity topology-encoded semantic anchors as the two sides of alignment.

3. We derive a metric-based fuzzy membership function from a variational formulation, providing a principled and unified alignment operator for modeling graded and non-exclusive diagnostic reasoning.

## 2. Related Work

### 2.1. Multi-Label Diagnosis

The evolution of Multi-Label Diagnosis (MLD) has largely centered on enhancing feature expressiveness and modeling label correlations. Early discriminative approaches prioritized global representation learning, which utilizes hybrid CNN-Transformer architectures (Liu et al., 2021; Wu et al., 2023; Zhang et al., 2023; Öztürk et al., 2025; Chen et al., 2025) to extract visual features, often augmented by Graph Neural Networks (GNNs) (Chen et al., 2019; 2020) to explicitly encode statistical label dependencies. More recently, the paradigm has shifted towards semantic alignment, leveraging Vision-Language (V-L) foundation models to introduce textual supervision. Approaches such as Med-CLIP (Wang et al., 2022) and prompt-learning frameworks (Huang et al., 2024; Li et al., 2025) adopt dual-encoder architectures to align visual regions with disease descriptions via contrastive objectives, such as InfoNCE (Oord et al., 2018). Despite their architectural and training diversity, these paradigms share a fundamental limitation: they implicitly rely on *rigid discriminative objectives*. Discriminative classifiers assume geometric separability in Euclidean space (Nickel & Kiela, 2017; Liu et al., 2017; Su et al., 2026), while contrastive learning enforces instance-level mutual exclusivity by pulling positive pairs together and pushing all other concepts apart. In the context of medical comorbidities, this *hard alignment* mechanism induces gradient conflicts (Wang et al., 2022; Zhang et al., 2022; Yu et al., 2020), which inadvertently penalizes the inherent semantic overlap between biologically correlated diseases (e.g., hypertension and diabetes). Moreover, it fails to account for visual polysemy, where a single pathological primitive supports multiple diagnostic interpretations.

## 2.2. Deep Fuzzy Learning

Fuzzy set theory, originally introduced by Zadeh (Zadeh, 1965), provides a mathematical framework for modeling graded and non-exclusive membership, and has been widely adopted to handle ambiguity and uncertainty in complex systems. We first introduce the critical definitions of fuzzy set theory:

**Definition 2.1.** A universe $\mathbb{U}$ is the set of all possible elements in a given domain of discourse.

**Definition 2.2.** A membership function $\mu : \mathbb{U} \to [0, 1]$ assigns each element a degree of membership.

**Definition 2.3.** A fuzzy set $\mathcal{A}$ on $\mathbb{U}$ is defined by its membership function $\mu$, i.e., $\mathcal{A} = \{(u, \mu(u)) \mid u \in \mathbb{U}\}$.

Recent works have integrated fuzzy logic into deep neural networks for segmentation and classification (Cao et al., 2024; Gu et al., 2022; Flügel et al., 2024), typically through a *fuzzy–defuzzy* pipeline where fuzzy layers act as intermediate feature refiners. However, in these methods, fuzziness is treated as an internal regularization technique. The learning objective ultimately reverts to rigid crisp decisions (defuzzification) before calculating the loss (e.g., Cross-Entropy), thereby severing the link to fuzzy theory during the critical gradient optimization phase. In contrast, FACT represents a paradigm shift from fuzzy representations to fuzzy objectives. Instead of ad-hoc fuzzy layers, we fundamentally reformulate the diagnosis task using a variational fuzzy set formulation. By deriving a distance-based fuzzy membership function as the learning objective itself, FACT enables direct optimization of graded, non-exclusive alignments between atomic visual evidence and semantic anchors. This design preserves the continuous and overlapping nature of disease semantics throughout the entire learning process, rather than collapsing it into discrete decisions at training time.

## 3. Method

As illustrated in Figure 2, FACT decomposes MLD into three complementary components. First, MLD is reformulated as a fuzzy alignment problem. Second, atomic visual evidence is constructed to capture visual polysemy. Third, semantic anchors are built to encode comorbidity topology priors.

### 3.1. Problem Reformulation

Let $(x, y)$ denote a data pair where $x \in \mathcal{X}$ is a medical image and $y \in \{0, 1\}^C$ represents the binary labels for $C$ diseases. Conventional MLD methods typically encode an image into a global representation and apply independent linear classifiers with sigmoid activations. Such approaches fundamentally adopt a discriminative boundary-based paradigm, implicitly assuming that disease categories can be separated by hyperplanes in a shared embedding space. In this work, we reformulate MLD as a metric-based fuzzy alignment problem between the atomic visual evidence within the image and the semantic anchor of that disease. This reformulation is guided by the principle that medical diagnosis is inherently evidence-driven rather than boundary-driven, where a disease is diagnosed by the presence and strength of supportive visual evidence, rather than by a global discriminative decision.

**Definition 3.1.** Let $\mathcal{V} = \{v_m\}_{m=1}^{M}$ be a set of unordered atomic visual evidence and $\mathcal{S} = \{s_c\}_{c=1}^{C}$ be a set of semantic anchors, where $\mathcal{V} \in \mathbb{R}^{M \times d}$ and $\mathcal{S} \in \mathbb{R}^{C \times d}$. $M$ denotes the number of atomic visual evidence, $C$ represents the total number of disease categories, and $d$ is the feature dimension. For each disease $c$, the diagnostic process is formulated as learning a fuzzy membership function $\mu_c = f(\mathcal{V}, s_c) \in [0, 1]$, which quantifies the graded compatibility between the visual evidence and the $c$-th disease concept.

Following Definition 3.1, our next objective is to explicitly construct the fuzzy membership functions $f: \mu_c = f(\mathcal{V}, s_c)$. However, it faces the challenge that an unordered set of visual atoms cannot be directly matched to a semantic point. To bridge this gap, a semantic interface operator is proposed to satisfy three desiderata: (i) permutation invariance over visual atoms, (ii) disease-specific evidence aggregation, and (iii) non-exclusive association to allow visual polysemy.

**Definition 3.2.** A semantic interface operator $\mathcal{I}$ that maps the unordered visual set to a collection of disease-aligned representations can be defined as:

$$E = \mathcal{I}(\mathcal{V}) = Q + \text{Sigmoid}\left(\frac{Q\mathcal{V}^{\top}}{\sqrt{d}}\right)\mathcal{V}, \qquad (1)$$

where $Q \in \mathbb{R}^{C \times d}$ denotes a set of learnable semantic matrices. Each row $e_c \in \mathbb{R}^d$ of $E$ encodes the aggregated visual evidence for the $c$-th disease. The operator $\mathcal{I}$ is naturally permutation-invariant, and its non-exclusive behavior is enabled by the sigmoid activation.

Thus, the original fuzzy membership function $\mu_c = f(\mathcal{V}, s_c)$ can be refactorized as:

$$\mu_c = f(\mathcal{I}_c(\mathcal{V}), s_c) = f(e_c, s_c), \qquad (2)$$

where $e_c$ and $s_c$ now reside in the same metric space $\mathbb{R}^d$.

**Theorem 3.3.** To ensure geometric continuity, we follow (Wahba, 2002) and model the membership function $f$ as a variational problem in a Reproducing Kernel Hilbert Space (RKHS):

$$\mathcal{E}[f] = \sum_{c=1}^{C} \left(y_c - f(e_c, s_c)\right)^2 + \lambda\|f\|_{\mathcal{H}_K}^2, \qquad (3)$$

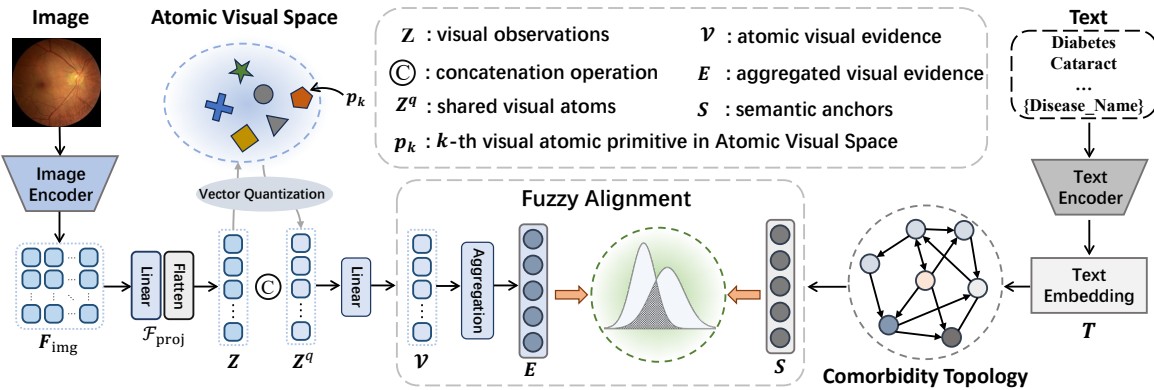

*Figure 2.* Overview of our proposed FACT framework. The pipeline comprises three integral components: (Left) Atomic Visual Evidence Construction, (Center) Metric-based Fuzzy Alignment, and (Right) Comorbidity Topology-encoded Semantic Anchors.

where $(e_c, s_c)$ denote the aggregated visual evidence and the semantic anchor of the $c$-th disease, $\mathcal{H}$ is the RKHS, and $K$ is the spatial kernel.

According to the Representer Theorem (Kimeldorf & Wahba, 1971), the minimizer admits a kernel expansion, and requiring smoothness naturally leads to a translation-invariant kernel with exponentially decaying spectral weight $\hat{K}(\omega) \propto \exp(-\sigma^2\|\omega\|^2/2)$ in the Fourier domain. Applying the inverse transform yields a Gaussian kernel in space:

$$f(e_c, s_c) \propto \exp\Big(-\frac{\|e_c - s_c\|^2}{2\sigma^2}\Big). \qquad (4)$$

Let $\tau = 2\sigma^2$, the fuzzy membership is finally defined as

$$\mu_c = \exp\Big(-\frac{\|e_c - s_c\|^2}{\tau}\Big), \qquad (5)$$

which decays smoothly as the evidence moves away from the semantic anchor.

The detailed derivation process can be found in the Appendix A.1, and the Lipschitz continuity analysis is provided in the Appendix A.2.

### 3.2. Atomic Visual Evidence Construction

To instantiate the metric alignment defined in Definition 3.1, the visual representation must be decomposable into discrete, reusable primitives rather than entangled global features. Given the input image $x$, a learnable backbone $\mathcal{F}_{\text{encoder}}$ is utilized to extract the dense feature map $\boldsymbol{F}_{\text{img}} \in \mathbb{R}^{D \times H \times W}$. To align with the metric embedding dimension $d$, we flatten the spatial dimensions and project the features to obtain $\boldsymbol{Z} = \mathcal{F}_{\text{proj}}(\boldsymbol{F}_{\text{img}}) \in \mathbb{R}^{M \times d}$, where $M = H \times W$ denotes the number of visual patches, and $\boldsymbol{Z} = \{z_m\}_{m=1}^M$ represents the visual observations.

Next, we employ a Vector Quantization (VQ) mechanism to decompose them into shared reusable visual atomic primitives. The Atomic Visual Space can be defined as a discrete

set $\mathcal{P} = \{p_k\}_{k=1}^K \subset \mathbb{R}^d$, where each $p_k$ represents a learnable visual atomic primitive. Each continuous token $z_m$ is mapped to its nearest topological neighbor in $\mathcal{P}$:

$$z_m^q = p_k, \quad \text{where } k = \underset{j\in\{1,\dots,K\}}{\arg\min} \|z_m - p_j\|_2. \quad (6)$$

The shared visual atoms $z_m^q$ can be stacked to get the $Z^q = \{z_m^q\}_{m=1}^M$. To preserve gradient flow for fine-grained details, we fuse the $Z^q$ with the original residuals via a linear projection $\mathcal{V} = \text{Linear}(Z \oplus Z^q)$, to get the atomic visual evidence $\mathcal{V} = \{v_m\}_{m=1}^M$.

### 3.3. Semantic Anchors Construction

Unlike orthogonal word embeddings, we construct the semantic anchors $\mathcal{S}$ that explicitly encode the comorbidity topology to capture pathological dependencies. First, we construct a prior Comorbidity Graph $\mathcal{G}$ where nodes represent disease categories. The adjacency matrix $\mathbf{A} \in \mathbb{R}^{C \times C}$ is initialized based on the conditional co-occurrence ratio derived from the clinical data. To encode this topological prior into the semantic representation space, we employ a 2-layer Graph Convolutional Network (GCN) to propagate semantic context among anchors. Let $T \in \mathbb{R}^{C \times d}$ denote the text embeddings of disease names, which are extracted from a text encoder. The initial node features are set as $\mathbf{S}^{(0)} = T$. The propagation rule for the $l$-th layer is defined as:

$$\mathbf{S}^{(l+1)} = \text{RELU}\Big(\tilde{\mathbf{D}}^{-\frac{1}{2}}\tilde{\mathbf{A}}\tilde{\mathbf{D}}^{-\frac{1}{2}}\mathbf{S}^{(l)}\mathbf{W}^{(l)}\Big). \qquad (7)$$

where $\tilde{\mathbf{A}} = \mathbf{A} + \mathbf{I}$ is the self-looped adjacency matrix, $\tilde{\mathbf{D}}$ is the degree matrix, and $\mathbf{W}$ is the weight matrix of GCN. Formally, we define the final set of semantic anchors as $\mathcal{S} = \{s_c\}_{c=1}^C$, where $s_c$ corresponds to the $c$-th row of $\mathbf{S}^{(L)}$. The details can be found in Appendix B.1.

### 3.4. Optimization Objective

The objective function is designed to jointly optimize the metric-based fuzzy alignment and the discrete structure of

Atomic Visual Evidence.

### 3.4.1. FUZZY ALIGNMENT LOSS

Standard BCE treats diseases independently, causing the model to be overwhelmed by easy negatives while under-penalizing hard errors. To address these limitations, we propose a composite Fuzzy Alignment Loss, including a point-level fuzzy loss formulated via a Log-Sum-Exp operator to prioritize hard errors, and a set-level fuzzy loss to explicitly enforce structural consistency over overlapping disease grounded in Łukasiewicz fuzzy logic. A detailed analysis against standard BCE can be found in Appendix A.3.1.

**Proposition 3.4.** Given membership scores $\mu_c \in (0, 1)$ and binary labels $y_c \in \{0, 1\}$, the disease-wise error contribution can be defined as:

$$r_c = \begin{cases} \frac{\mu_c}{1-\mu_c}, & y_c = 0, \\ \frac{1-\mu_c}{\mu_c}, & y_c = 1. \end{cases} \quad (8)$$

The point-level fuzzy loss is then formulated as:

$$\mathcal{L}_{\text{point}} = \log\left(1 + \sum_{c:y_c=0} r_c\right) + \log\left(1 + \sum_{c:y_c=1} r_c\right). \quad (9)$$

**Proposition 3.5.** To explicitly quantify the overlap of co-occurrence diseases, we adopt Łukasiewicz fuzzy logic, which provides a principle for fuzzy conjunction (AND) and disjunction (OR) in a continuous domain:

$$\begin{aligned} T_{Luk}(\mu_c, y_c) &= \max(0, \mu_c + y_c - 1), \\ S_{Luk}(\mu_c, y_c) &= \min(1, \mu_c + y_c), \end{aligned} \quad (10)$$

where $T_{Luk}$ and $S_{Luk}$ denote the t-norm (intersection) and t-conorm (union), respectively. Based on these operators, the Fuzzy Jaccard Similarity is formulated to measure the set-level overlap:

$$\mathcal{J}(\boldsymbol{\mu}, \mathbf{y}) = \frac{\sum_{c=1}^{C} T_{Luk}(\mu_c, y_c)}{\sum_{c=1}^{C} S_{Luk}(\mu_c, y_c) + \epsilon}. \quad (11)$$

Then, the set-level fuzzy loss can be defined as:

$$\mathcal{L}_{\text{set}} = 1 - \mathcal{J}(\boldsymbol{\mu}, \mathbf{y}). \quad (12)$$

The total Fuzzy Alignment Loss can be finally expressed as:

$$\mathcal{L}_{\text{fuzzy}} = \mathcal{L}_{\text{point}}(\boldsymbol{\mu}, \mathbf{y}) + \alpha \cdot \mathcal{L}_{\text{set}}(\boldsymbol{\mu}, \mathbf{y}). \quad (13)$$

The detailed gradient analysis and convexity analysis can be found in Appendix A.3.2 and Appendix A.3.3.

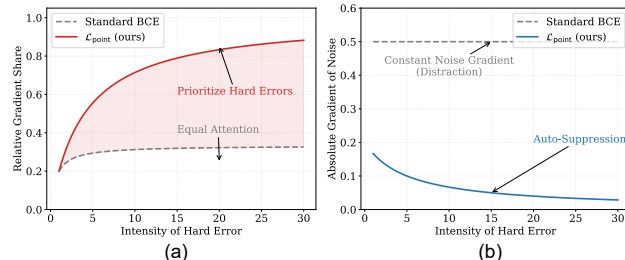

*Figure 3.* Gradient Dynamics Analysis. We simulate an optimization landscape containing one emerging hard error and fixed background noise.

### 3.4.2. ATOMIC VISUAL COMMITMENT LOSS

To learn the Atomic Visual Space $\mathcal{P}$ and update the encoder, we adopt the standard commitment loss in VQ:

$$\mathcal{L}_{\text{commit}} = \|\text{sg}[\boldsymbol{Z}] - \boldsymbol{Z}^q\|_2^2 + \beta\|\boldsymbol{Z} - \text{sg}[\boldsymbol{Z}^q]\|_2^2, \quad (14)$$

where $\boldsymbol{Z}^q$ are the quantized tokens, $\text{sg}[\cdot]$ denotes the stop-gradient operator, and $\beta = 0.25$ is the hyperparameter balancing the learning rates of the atomic visual space and the encoder following original VQ works (Van Den Oord et al., 2017; Esser et al., 2021; Huang et al., 2026).

### 3.4.3. TOTAL LOSS AND ANALYSIS

The final optimization objective is a weighted summation of the fuzzy alignment terms and the representation constraint:

$$\mathcal{L}_{\text{total}} = \mathcal{L}_{\text{fuzzy}} + \gamma \cdot \mathcal{L}_{commit}. \quad (15)$$

**Anaylsis** Figure 3 analyzes gradient behavior under heterogeneous error magnitudes. As shown in Figure 3(a), unlike standard BCE, which distributes gradients linearly across errors, the proposed fuzzy loss adaptively concentrates gradient mass on dominant hard errors, enabling rapid corrective updates. Moreover, Figure 3(b) shows effective noise suppression. As the primary error increases, the gradients induced by background noise are progressively attenuated, whereas BCE assigns persistent gradients to irrelevant errors.

## 4. Experiments

### 4.1. Experimental Settings

**Datasets** We evaluate FACT on three publicly available MLD benchmarks spanning different anatomical regions and data regimes, including ODIR-5K (Li et al., 2021), NIH-Chest (Wang et al., 2017), and RSNA-IHD (Flanders et al., 2020). This diverse selection allows us to assess robustness across different modalities and scales. To further examine robustness under severe class imbalance and noisy clinical annotations, we additionally conduct experiments on

*Table 1.* The quantitative comparisons with SOTA MLD methods on the ODIR-5K, NIH-Chest, and RSNA-IHD datasets. The optimal result is represented in bold, the second-best result is underlined.

| Methods | ODIR-5K | | | | NIH-Chest | | | | RSNA-IHD | | | |
|---|---|---|---|---|---|---|---|---|---|---|---|---|
| | mAP↑ | F1↑ | AUC↑ | Avg.↑ | mAP↑ | F1↑ | AUC↑ | Avg.↑ | mAP↑ | F1↑ | AUC↑ | Avg.↑ |
| TransferNet (2021) | 64.30 | 86.29 | 89.01 | 79.87 | 51.71 | 90.12 | 86.73 | 76.19 | 84.53 | 90.36 | 95.44 | 90.11 |
| ASL (2021) | 59.69 | 87.84 | 90.41 | 79.31 | 42.32 | 86.31 | 85.41 | 71.35 | 83.07 | 91.97 | 96.15 | 90.40 |
| RAL (2023) | 59.61 | 86.86 | 89.77 | 78.75 | 38.63 | 83.51 | 83.78 | 68.64 | 82.91 | 92.02 | 96.24 | 90.39 |
| TADCL (2023) | 65.04 | 86.57 | 86.45 | 79.35 | 41.05 | 86.91 | 84.22 | 70.73 | 87.47 | 92.07 | 96.32 | 91.95 |
| Two-Way (2023) | 55.93 | 84.13 | 91.17 | 77.08 | 46.38 | 84.81 | 86.70 | 72.63 | 57.51 | 79.17 | 94.37 | 77.02 |
| CTransCNN (2023) | 69.30 | 90.29 | 91.35 | 83.65 | 52.40 | 87.88 | 85.61 | 75.30 | 85.99 | 90.45 | 94.69 | 90.38 |
| LDR (2024) | 69.69 | 89.79 | 90.81 | 83.43 | 57.81 | 90.01 | 87.52 | 78.45 | 84.85 | 89.41 | 94.54 | 89.60 |
| SupCon (2024) | 67.43 | 86.75 | 90.62 | 81.60 | 58.82 | 89.28 | 86.08 | 78.06 | 84.59 | 90.03 | 95.12 | 89.91 |
| MultiCo (2025) | 68.30 | 89.14 | 91.45 | 82.96 | 54.32 | 90.14 | 87.04 | 77.17 | 86.70 | 92.11 | 96.22 | 91.68 |
| HydraViT (2025) | 67.92 | 88.25 | 90.22 | 82.13 | 54.60 | 90.33 | 86.96 | 77.30 | 86.39 | 91.22 | 96.33 | 91.31 |
| FACT (Ours) | **72.61** | **90.91** | **92.97** | **85.50** | **58.95** | **90.44** | **87.81** | **79.07** | **89.72** | **92.97** | **96.90** | **93.20** |
| Δ | +2.92 | +0.62 | +1.52 | +1.85 | +0.13 | +0.11 | +0.29 | +0.62 | +2.25 | +0.86 | +0.57 | +1.25 |

*"△" represents the difference in metrics between our method and the second-best method.*

CXR-LT (Holste et al., 2025), a large-scale long-tailed chest X-ray benchmark with over 370K images and 40 disease categories. Additional dataset-specific details are provided in Appendix C.1.

**Baselines and Metrics** We compare FACT with a diverse suite of competitive baselines published in Top-Tier conference and journals that cover 10 mainstream MLD paradigms, including TransferNet (Gour & Khanna, 2021), ASL (Ridnik et al., 2021), RAL (Park et al., 2023), TADCL (Zhang et al., 2023), Two-Way (Kobayashi, 2023), CTransCNN (Wu et al., 2023), LDR (2024), SupCon (Zhang & Wu, 2024), MultiCo (Xu et al., 2025), HydraViT (Öztürk et al., 2025). The detailed information of competitive baselines can be found in Appendix C.2. Regarding evaluation metrics, we follow the standard evaluation protocols of the respective benchmarks (Li et al., 2021; Wang et al., 2017; Flanders et al., 2020) and report mean Average Precision (mAP), micro F1-score (F1), and Area Under the Curve (AUC) across all datasets. Unless otherwise specified, F1 is computed with a threshold of 0.5 for all pathologies.

**Implementation Details** We implement FACT using the PyTorch framework (Paszke et al., 2019) and conduct all experiments on NVIDIA GeForce RTX 4090 GPUs. Input images are processed to $224 \times 224$ with horizontal and vertical augmentations during training. The model is optimized using the Adam optimizer (Kingma, 2014) with a learning rate of $1 \times 10^{-4}$ for 100 epochs. Regarding hyperparameters, the fuzzy kernel temperature is fixed at $\tau = 0.5$. We perform strictly patient-level stratification and randomly split the data into 80% training and 20% testing, and the random seed is fixed at 1234 to ensure reproducibility.

### 4.2. Main Results

Table 1 demonstrates that FACT consistently achieves state-of-the-art performance across all benchmarks, outperforming diverse baselines in every reported metric and modality. A clear trend in the data reveals that while competitive baselines achieve respectable AUC scores, their mAP and F1-scores fluctuate significantly across datasets. For instance, LDR performs strongly on ODIR-5K but fails to maintain its advantage on RSNA-IHD, while TADCL shows the opposite behavior. This inconsistency reveals a fundamental lack of robustness in current methods when navigating varying comorbidity structures and data scales. Architecture-centric methods like HydraViT and CTransCNN still rely on independent binary relevance, which enforces a geometric rigidity that treats concurrent diseases as isolated decision planes, thereby failing to resolve overlapping visual evidence. Meanwhile, objective-centric and semantic-guided methods such as ASL and SupCon suffer from a contrastive exclusivity bias in optimization, inadvertently suppressing shared pathological primitives and penalizes the natural semantic overlap between biologically correlated diseases, leading to suboptimal mAP on complex datasets like ODIR-5K and RSNA-IHD. In contrast, the significant gains of 2.92% and 2.25% in mAP on ODIR and RSNA respectively, demonstrate that our proposed FACT provides a superior framework for capturing the non-orthogonal semantics inherent in clinical diagnosis.

### 4.3. Disease-wise Performance

To provide a granular assessment of FACT across diverse pathological manifestations, Figure 4 reports the per-class AUC for all 27 diseases spanning the three evaluated datasets. As shown in the radar chart, FACT (red) forms a consistently strong outer envelope that largely encompasses

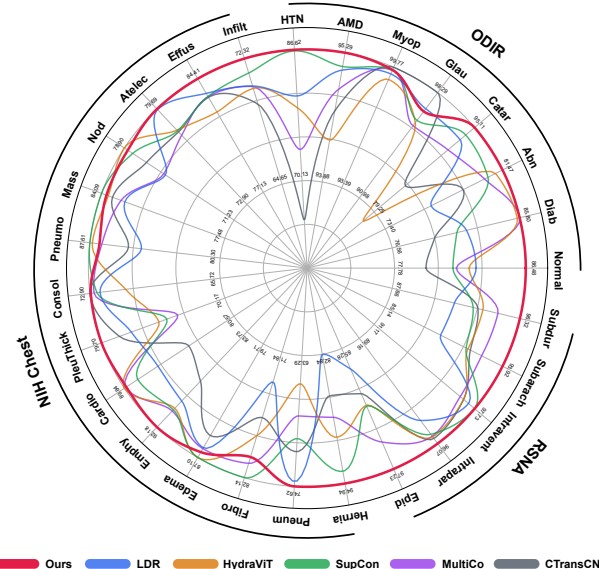

*Figure 4.* Comprehensive evaluation of AUC performance across all 27 diseases on the ODIR, NIH-Chest, RSNA dataset. The plotting range for each category is normalized to [Min - 10, Max] (in %), where Min and Max denote the lowest and highest AUC scores among compared methods.

the profiles of the top-5 state-of-the-art baselines, indicating that the gains are broadly distributed across categories rather than driven by a small subset of diseases. Detailed results for all baselines are provided in Appendix D.

*Table 2.* Ablation study investigating the contribution of individual components in the FACT. ① denotes the Atomic Visual Space, and ② represents the Comorbidity Topology.

| No. | ① | ② | mAP↑ | F1↑ | AUC↑ | Avg.↑ |
|-----|---|---|------|-----|------|-------|
| #1 | ✗ | ✗ | 60.73 | 88.38 | 88.79 | 79.30 |
| #2 | ✓ | ✗ | 70.31 | 90.00 | 92.62 | 84.31 |
| #3 | ✗ | ✓ | 69.12 | 90.70 | 92.42 | 84.08 |
| #4 | ✓ | ✓ | **72.61** | **90.91** | **92.97** | **85.50** |

*Table 3.* Ablation study investigating the impact of loss functions in the FACT.

| No. | $\mathcal{L}_{point}$ | $\mathcal{L}_{set}$ | $\mathcal{L}_{commit}$ | mAP↑ | F1↑ | AUC↑ | Avg.↑ |
|-----|-------|------|---------|------|-----|------|-------|
| #1 | ✓ | ✗ | ✗ | 68.91 | 89.00 | 92.02 | 83.31 |
| #2 | ✓ | ✓ | ✗ | 69.12 | 90.70 | 92.42 | 84.08 |
| #3 | ✓ | ✗ | ✓ | 72.11 | 90.61 | 92.20 | 84.97 |
| #4 | ✓ | ✓ | ✓ | **72.61** | **90.91** | **92.97** | **85.50** |

## 4.4. Ablation Study

To assess the effectiveness of FACT's design, we conduct a component-wise ablation on the ODIR-5K dataset (Table 2). Applying fuzzy alignment directly to entangled global features yields limited performance, highlighting the difficulty of aligning ambiguous representations without structural

decomposition. Introducing the Atomic Visual Space leads to a substantial improvement, validating its role in resolving visual polysemy via shared and reusable primitives. Incorporating the Comorbidity Topology also provides clear gains, confirming the importance of topology-informed semantic anchors for leveraging disease correlations. The full model achieves the best results across all metrics, demonstrating the complementary contributions of visual decomposition and semantic structuring to effective fuzzy alignment.

Table 3 analyzes the contributions of different optimization objectives. Using only the Fuzzy Point Alignment Loss ($\mathcal{L}_{point}$) provides basic discriminability but yields a limited mAP of 68.91%. Adding the Fuzzy Set Jaccard Loss ($\mathcal{L}_{set}$) improves the F1 by 1.70%, indicating that set-level supervision enhances global label consistency beyond point-wise alignment. The VQ Commitment Loss ($\mathcal{L}_{commit}$) is critical for feature quality, boosting mAP to 72.11% by stabilizing the atomic visual space and preserving meaningful metric structure. The full objective achieves the strongest and most stable performance, confirming the complementary roles of point-level alignment, set-level consistency, and metric regularization.

*Table 4.* Quantitative analysis of the effect between the alignment operator and the optimization objective. We compare the Fuzzy strategy against the Rigid baseline under different loss constraints.

| No. | Align. | | Objec. | | Metrics | | | | |
|-----|-------|-------|---------------|--------------|-------|------|------|------|------|
| | Fuzzy | Rigid | $\mathcal{L}_{fuzzy}$ | $\mathcal{L}_{BCE}$ | mAP↑ | F1↑ | AUC↑ | CER↓ | HAM↓ |
| #1 | ✓ | ✗ | ✓ | ✗ | **72.61** | **90.91** | **92.97** | **16.22** | **9.82** |
| #2 | ✗ | ✓ | ✗ | ✓ | 72.22 | 90.90 | 92.24 | 27.51 | 10.25 |
| #3 | ✓ | ✗ | ✗ | ✓ | 71.84 | 90.50 | 91.86 | 31.08 | 10.45 |

## 4.5. Analysis of Fuzzy Alignment

Table 4 compares the proposed metric-based fuzzy alignment with the conventional rigid discriminative paradigm. FACT (Row #1) strictly dominates the Rigid baseline (Row #2) across all metrics, with particularly pronounced gains in reliability. The Calibration Error Rate is nearly halved ($27.51 \rightarrow 16.22$), indicating that rigid partitioning suffers from severe overconfidence despite reasonable discriminative performance. In contrast, FACT produces predictions that are better calibrated to underlying pathological likelihoods. Row #3 reveals a critical coupling between the alignment operator and the optimization objective. When the Gaussian membership function is optimized with standard BCE loss, performance collapses, yielding the lowest mAP and the worst calibration. This confirms that fuzzy alignment must be paired with a compatible metric-based objective to be effective.

## 4.6. Analysis of Atomic Visual Space

To verify whether FACT learns semantically meaningful and reusable visual primitives, Figure 5 visualizes the acti-

*Table 5.* Quantitative comparison of different topology initialization strategies for the comorbidity graph.

| Variant | ODIR-5K | | | | NIH-Chest | | | | RSNA-IHD | | | |
|---|---|---|---|---|---|---|---|---|---|---|---|---|
| | mAP↑ | F1↑ | AUC↑ | Avg.↑ | mAP↑ | F1↑ | AUC↑ | Avg.↑ | mAP↑ | F1↑ | AUC↑ | Avg.↑ |
| Identity | 71.63 | 90.75 | 92.83 | 85.07 | 58.83 | 90.26 | 87.70 | 78.93 | 87.11 | 91.41 | 95.91 | 91.48 |
| Shuffled | 70.55 | 89.38 | 92.14 | 84.02 | 58.81 | 90.31 | 87.34 | 78.82 | 89.04 | 91.92 | 96.06 | 92.34 |
| Random | 71.56 | 89.21 | 91.77 | 84.18 | 58.85 | 90.29 | 87.39 | 78.84 | 88.83 | 91.89 | 96.00 | 92.24 |
| Ours | **72.61** | **90.91** | **92.97** | **85.50** | **58.95** | **90.44** | **87.81** | **79.07** | **89.72** | **92.57** | **96.90** | **93.20** |

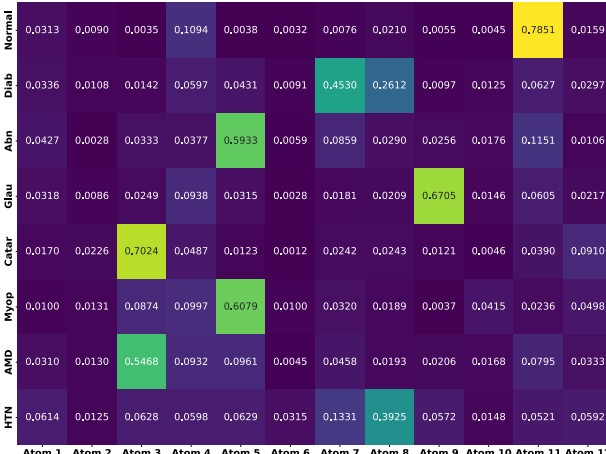

*Figure 5.* Visualization of the activation affinity between disease categories and visual atoms.

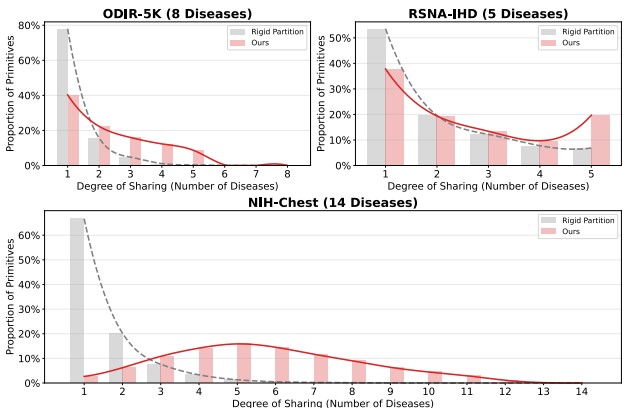

*Figure 6.* Distribution of atomic visual evidence. The horizontal axis represents the degree of sharing, defined as the count of disease categories activating a specific visual primitive, while the vertical axis denotes the proportion of such primitives within the learned Atomic Visual Space.

vation affinity between disease categories and atomic visual evidence. The heatmap exhibits sparse and structured selectivity, where distinct pathologies preferentially attend to specific atoms (e.g., Normal–Atom 11, Glaucoma–Atom 9). More importantly, it reveals clear visual polysemy as single atoms are jointly activated by correlated diseases. For instance, Atom 3 is co-activated by Cataract and AMD, while Atom 8 simultaneously supports Diabetes and Hypertension,

indicating that FACT enables shared visual evidence rather than enforcing feature exclusivity.

Figure 6 further quantifies this behavior by comparing primitive usage distributions with a rigid partition baseline. The baseline predominantly assigns visual features to a single disease, reflecting strong geometric rigidity. In contrast, FACT shifts the distribution toward higher degrees of sharing. This effect is most pronounced on RSNA-IHD, where many primitives are activated across all hemorrhage subtypes, and on NIH-Chest, where usage peaks around 4–5 diseases, consistent with known comorbidity patterns. These results confirm that FACT learns an intrinsically polysemous visual vocabulary, effectively breaking the feature exclusivity imposed by conventional MLD models.

### 4.7. Analysis of Comorbidity Topology

To assess the sensitivity of FACT to semantic priors, we evaluate the impact of different graph topology initializations in Table 5. The proposed clinical prior consistently yields the highest performance across all three datasets. For instance, on the RSNA-IHD dataset, the mAP decreases from 89.72% to 87.11% when switching to the Identity matrix. This decline indicates that modeling diseases as independent entities fails to capture the underlying pathological correlations. Furthermore, the Shuffled and Random variants frequently perform worse than the Identity baseline. This observation confirms that incorporating incorrect or noisy dependencies harms the feature alignment process. Consequently, accurate medical prior knowledge is essential for effective multi-label diagnosis.

### 4.8. Analysis of Fuzzy Membership Kernel

We further analyze the quantitative effect of different fuzzy membership kernels within FACT, aiming to validate our argument that the Gaussian kernel provides a suitable membership form under the requirements of membership continuity and semantic consistency. As reported in Table 6, we replace the Gaussian membership with several alternative kernels, including linear, cosine, Laplacian, and Polynomial kernels. The results show that replacing the Gaussian membership consistently degrades performance across datasets. This demonstrates that the derived fuzzy membership formulation instantiated with the Gaussian kernel is effective

*Table 6.* Quantitative comparison of different fuzzy membership kernels.

| Kernels | ODIR-5K | | | | NIH-Chest | | | | RSNA-IHD | | | |
|---|---|---|---|---|---|---|---|---|---|---|---|---|
| | mAP↑ | F1↑ | AUC↑ | Avg.↑ | mAP↑ | F1↑ | AUC↑ | Avg.↑ | mAP↑ | F1↑ | AUC↑ | Avg.↑ |
| Linear | 69.94 | 89.64 | 91.82 | 83.80 | 51.94 | 89.85 | 87.23 | 76.34 | 89.15 | 92.16 | 96.64 | 92.65 |
| Cosine | 68.93 | 89.66 | 91.66 | 83.42 | 52.69 | 90.15 | 87.45 | 76.76 | 89.14 | 92.03 | 96.65 | 92.61 |
| Laplacian | 30.98 | 85.36 | 76.20 | 64.18 | 27.63 | 88.70 | 76.04 | 64.12 | 48.66 | 74.80 | 73.15 | 65.54 |
| Polynomial | 33.85 | 14.64 | 76.47 | 41.65 | 35.76 | 12.58 | 75.17 | 41.17 | 86.97 | 28.46 | 91.22 | 68.88 |
| Ours | **72.61** | **90.91** | **92.97** | **85.50** | **58.95** | **90.44** | **87.81** | **79.07** | **89.72** | **92.97** | **96.90** | **93.20** |

*Table 7.* Quantitative reliability analysis comparing Hamming Loss (HAM) and Confident Error Rate (CER) across three datasets.

| Methods | ODIR-5K | | NIH-Chest | | RSNA-IHD | |
|---|---|---|---|---|---|---|
| | HAM↓ | CER↓ | HAM↓ | CER↓ | HAM↓ | CER↓ |
| LDR(2024) | 9.98 | 29.78 | 9.69 | 22.19 | 7.61 | 16.58 |
| MultiCo(2025) | 10.86 | 19.39 | 6.72 | 18.33 | 8.32 | 18.39 |
| HydraViT(2025) | 11.30 | 26.54 | 6.61 | 18.43 | 8.11 | 15.34 |
| Ours | **9.82** | **16.22** | **6.42** | **14.48** | **7.42** | **9.09** |

and empirically favorable.

## 4.9. Evaluation under Long-Tailed and Noisy Settings

To further evaluate robustness under severe class imbalance and noisy clinical annotations, we conduct additional experiments on CXR-LT, a large-scale long-tailed chest X-ray benchmark with over 370K images and 40 disease categories. For Task 1, both training and test labels are automatically extracted from radiology reports, forming a noisy-train/noisy-test setting with 40 disease labels. As shown in Table 8, FACT achieves competitive performance on Head classes and obtains the best results on Middle and Tail classes, demonstrating its effectiveness under large-scale noisy long-tailed evaluation. For Task 2, models are trained on report-extracted noisy labels and evaluated on a manually annotated gold-standard test set with 26 disease categories. As reported in Table 9, the full FACT design with comorbidity topology and the fuzzy objective performs best on Middle and Tail classes, indicating improved robustness when generalizing from noisy long-tailed training data to clean clinical annotations.

*Table 8.* Comparison on CXR-LT Task 1, a long-tailed noisy-train/noisy-test setting with 40 disease labels. Disease categories are grouped into Head, Middle, and Tail classes according to label prevalence.

| Methods | Groups | | |
|---|---|---|---|
| | Head | Middle | Tail |
| LDR (2024) | 77.30 | 74.75 | 75.73 |
| MultiCo (2025) | 76.84 | 70.06 | 64.71 |
| HydraViT (2025) | **78.15** | 76.02 | 77.66 |
| Ours | 77.88 | **78.41** | **81.20** |

*Table 9.* Ablation study on CXR-LT Task 2, a noisy-train/clean-test setting with 26 disease labels. Disease categories are grouped into Head, Middle, and Tail classes according to label prevalence. "Topo." denotes Comorbidity Topology and "Objec." denotes the training objective.

| No. | Topo. | | Objec. | | Groups | | |
|---|---|---|---|---|---|---|---|
| | w | w/o | $\mathcal{L}_{fuzzy}$ | $\mathcal{L}_{BCE}$ | Head | Middle | Tail |
| #1 | ✓ | ✗ | ✗ | ✓ | **74.14** | 71.13 | 67.16 |
| #2 | ✗ | ✓ | ✓ | ✗ | 72.59 | 68.29 | 72.87 |
| #3 | ✓ | ✗ | ✓ | ✗ | 73.83 | **71.92** | **74.30** |

## 4.10. Reliability Study

We evaluate the predictive reliability of FACT against three competitive baselines as detailed in Table 7. FACT achieves the lowest error rates across all datasets for both metrics. On the ODIR-5K dataset, our method reduces the CER to 16.22%, surpassing the second-best MultiCo by over 3%. The improvement is most pronounced on the RSNA-IHD dataset, where FACT lowers the CER to 9.09%, significantly outperforming HydraViT at 15.34%. Similarly, FACT maintains the lowest Hamming Loss on all benchmarks. These results demonstrate that the fuzzy alignment strategy reduces overconfident incorrect predictions and improves the overall calibration of the diagnostic model.

## 5. Conclusion

In this paper, we revealed the fundamental limitation of conventional MLD paradigms, which typically formulate the task as a rigid discriminative partitioning problem. We argue that this approach is flawed as it implicitly assumes that overlapping pathologies can be entirely separated. To address this, we proposed Fuzzy Alignment with Comorbidity Topology (FACT). Grounded in the observations of intrinsic visual polysemy and semantic correlation, FACT reformulates MLD as a metric-based fuzzy alignment process between atomic visual evidence and disease semantic anchors enhanced with comorbidity structure. Extensive experiments on three clinical datasets and an additional long-tailed noisy benchmark demonstrate that FACT effectively resolves the ambiguity of co-occurring diseases, achieving superior diagnostic performance and reliability compared to state-of-the-art approaches.

## Acknowledgements

This work was supported in part by the National Natural Science Foundation of China under Grant 62271335, and in part by Sichuan Science and Technology Program under Grant 2025ZNSFSC0470.

## Impact Statement

This work aims to improve the reliability and interpretability of MLD by introducing a fuzzy alignment framework. By modeling visual polysemy through reusable atomic visual evidence, encoding disease dependencies via semantic anchors encoded with comorbidity topology, and adopting a metric-based fuzzy membership formulation, the proposed approach provides a more faithful representation of non-exclusive and correlated clinical findings. The study is conducted exclusively on publicly available and anonymized datasets, and does not involve human subjects or patient interaction. While medical diagnosis is a high-stakes application domain, this work is intended as a methodological investigation rather than a deployable clinical system. Any real-world use would require extensive validation and regulatory approval. We do not foresee specific adverse ethical or societal impacts beyond the general considerations associated with medical AI systems, such as the need for responsible deployment and human oversight.

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

## A. Theoretical Proofs

### A.1. Derivation of Membership Function

Recall from Theorem 4.3 that we aim to recover a function $f : \mathbb{R}^d \times \mathbb{R}^d \to \mathbb{R}$ that minimizes the empirical risk while maintaining maximum geometric smoothness. Note that the function $f$ optimized in the RKHS serves as an intermediate smooth function and is not restricted to $[0, 1]$ while its values are later mapped to define the fuzzy membership, which lies in $[0, 1]$. We formulate this as finding the minimizer $f^*$ in a Reproducing Kernel Hilbert Space (RKHS) $\mathcal{H}$:

$$f^* = \arg\min_{f \in \mathcal{H}} \mathcal{E}[f] = \underbrace{\sum_{c=1}^{C} (y_c - f(e_c, s_c))^2}_{\text{Data Fidelity}} + \lambda \underbrace{\int_{\mathbb{R}^d} |\hat{f}(\omega)|^2 \exp\left(\frac{\|\omega\|^2}{2}\right) d\omega}_{\text{Regularization}}, \tag{16}$$

where $\|f\|_{\mathcal{H}_K}^2 = \int_{\mathbb{R}^d} \frac{|\hat{f}(\omega)|^2}{\hat{K}(\omega)} d\omega$, $\lambda > 0$ is the regularization parameter, and $\hat{f}(\omega)$ is the Fourier transform of $f$. Here, $\omega \in \mathbb{R}^d$ corresponds to the frequency components along each input dimension of $f$.

To enforce an infinite-order smoothness constraint that ensures the function decays rapidly away from the semantic anchors, we define the smoothness penalty in the frequency domain. By Parseval's identity, the squared norm of $f$ in the RKHS $\mathcal{H}$ can be expressed as:

$$\int_{\mathbb{R}^d} |\hat{f}(\omega)|^2 \exp\left(\frac{\|\omega\|^2}{2}\right) d\omega = \int_{\mathbb{R}^d} \frac{|\hat{f}(\omega)|^2}{\hat{K}(\omega)} d\omega, \tag{17}$$

where $\hat{f}(\omega)$ denotes the Fourier transform of $f$, and $\hat{K}(\omega)$ is the spectral weight function controlling the penalization of each frequency component, with higher frequencies typically being penalized more strongly.

To enforce a smoothness property that produces a Gaussian profile in the spatial domain, we choose an exponentially decaying spectral weight in the frequency domain:

$$\hat{K}(\omega) = \exp\left(-\frac{\sigma^2 \|\omega\|^2}{2}\right), \tag{18}$$

where the parameter $\sigma^2$ controls the degree of smoothness. Larger $\sigma^2$ leads to stronger suppression of high-frequency components, resulting in a more rapidly decaying function in space, while smaller $\sigma^2$ produces a slower decay.

Substituting Eq. (18) into Eq. (17), the regularization term can be written explicitly as:

$$\Omega[f] = \int_{\mathbb{R}^d} |\hat{f}(\omega)|^2 \exp\left(\frac{\sigma^2 \|\omega\|^2}{2}\right) d\omega, \tag{19}$$

where $\Omega[f]$ denotes the smoothness penalty functional that quantifies the contribution of each frequency component to the overall regularization. This formulation heavily penalizes high-frequency components, thereby enforcing extreme smoothness in the learned function.

According to the Representer Theorem, the minimizer $f^*$ of the functional in Eq. (16) admits a representation as a finite linear combination of kernels centered at the semantic anchors $s_c$ and aggregated visual evidence $e_c$:

$$f^*(x) = \sum_j \phi_j K(u, u_j), \tag{20}$$

where the coefficients $\phi_j$ are determined by minimizing $\mathcal{J}[f]$. Although the RKHS is infinite-dimensional, the theorem guarantees that the minimizer lies in the span of kernels evaluated at the training points. Here, the reproducing kernel $K(u, u')$ can be formally interpreted as the Green's function corresponding to the regularization operator defined by $\hat{K}(\omega)$.

To obtain the spatial form of the kernel $K(u, u')$, we perform the inverse Fourier transform (IFT) of the spectral weight $\hat{K}(\omega)$, denoted by $\mathcal{F}^{-1}[\cdot]$. Since the kernel is translation-invariant, it depends only on the difference between inputs, so we define $K(u, u') = k(u - u')$, where $\hat{k}(\omega) = \hat{K}(\omega)$.

$$\mathcal{F}^{-1}\left[\exp\left(-\frac{\sigma^2 \|\omega\|^2}{2}\right)\right](u) = \frac{1}{(2\pi\sigma^2)^{d/2}} \exp\left(-\frac{\|u\|^2}{2\sigma^2}\right). \tag{21}$$

Thus, the spatial-domain kernel induced by the frequency-domain Gaussian weighting is itself a Gaussian function, with variance controlled by $\sigma^2$.

Let $u = e_c$ and $u' = s_c$. Up to a multiplicative constant, the kernel function is:

$$K(e_c, s_c) \propto \exp\left(-\frac{\|e_c - s_c\|^2}{2\sigma^2}\right). \tag{22}$$

In our formulation (Eq. (5) in the main text), we introduce a learnable temperature parameter $\tau$, identifying $\tau = 2\sigma^2$. This allows us to express the fuzzy membership as:

$$\mu_c = \exp\left(-\frac{\|e_c - s_c\|^2}{\tau}\right), \tag{23}$$

which corresponds exactly to the proposed fuzzy membership function.

### A.2. Lipschitz Continuity of Membership Function

For the Gaussian membership function defined in Eq. (5) with fixed anchor point $s_c$. The gradient with respect to $e_c$ is:

$$\nabla_e f(e_c) = -\frac{2}{\tau} f(e_c)(e_c - s_c), \tag{24}$$

where $\tau > 0$ is the temperature parameter controlling the rate of decay, and the gradient points in the direction that decreases the Euclidean distance between $e_c$ and $s_c$.

Using the chain rule, the gradient of $\mu_c$ with respect to $e_c$ is:

$$\nabla_{e_c} \mu_c = \nabla_{e_c} \exp\left(-\frac{\|e_c - s_c\|^2}{\tau}\right) = \exp\left(-\frac{\|e_c - s_c\|^2}{\tau}\right) \cdot \nabla_{e_c}\left(-\frac{\|e_c - s_c\|^2}{\tau}\right). \tag{25}$$

Since $\nabla_{e_c}\|e_c - s_c\|^2 = 2(e_c - s_c)$, we can obtain:

$$\nabla_{e_c} \mu_c = -\frac{2}{\tau} \mu_c (e_c - s_c). \tag{26}$$

Taking the Euclidean norm of the gradient gives:

$$\|\nabla_{e_c} \mu_c\| = \frac{2}{\tau} \mu_c \|e_c - s_c\| = \frac{2}{\tau} \|e_c - s_c\| \exp\left(-\frac{\|e_c - s_c\|^2}{\tau}\right). \tag{27}$$

Let $r = \|e_c - s_c\| \geq 0$, and define:

$$g(r) = \frac{2}{\tau} r \, e^{-r^2/\tau}. \tag{28}$$

We aim to find $\max_{r \geq 0} g(r)$ to obtain the Lipschitz constant. Differentiating $g(r)$ with respect to $r$:

$$g'(r) = \frac{2}{\tau}\left(e^{-r^2/\tau} - \frac{2r^2}{2\tau}e^{-r^2/\tau}\right) = \frac{2}{\tau}e^{-r^2/\tau}\left(1 - \frac{r^2}{\tau}\right). \tag{29}$$

Setting $g'(r) = 0$ gives

$$1 - \frac{r^2}{\tau} = 0 \quad \Rightarrow \quad r = \sqrt{\tau}. \tag{30}$$

The maximum gradient norm is therefore

$$\|\nabla_{e_c} \mu_c\|_{\max} = g(\sqrt{\tau}) = \frac{2}{\tau} \cdot \sqrt{\tau} \, e^{-\tau/\tau} = \frac{2}{e\sqrt{\tau}}. \tag{31}$$

Since the gradient norm is bounded, the Gaussian membership function is Lipschitz continuous:

$$\|\mu_c(e_1) - \mu_c(e_2)\| \leq L\|e_1 - e_2\|, \quad L = \frac{2}{e\sqrt{\tau}}. \tag{32}$$

**A.3. Proofs of fuzzy loss**

A.3.1. RELATIONSHIP WITH STANDARD BCE

The conventional BCE loss can be expressed as:

$$\mathcal{L}_{\text{BCE}} = -\sum_{c=1}^{C}\big[y_c \log \mu_c + (1 - y_c)\log(1 - \mu_c)\big] = \sum_{c=1}^{C}\log(1 + r_c). \tag{33}$$

In contrast, $\mathcal{L}_{\text{point}}$ first aggregates error contributions within each label group and then applies the logarithm. By the concavity of $\log(1 + \cdot)$ and Jensen's inequality, we have $\mathcal{L}_{\text{point}} \leq \mathcal{L}_{\text{BCE}}$ with equality if and only if all $r_c$ within each group are equal. This inequality indicates that $\mathcal{L}_{\text{point}}$ encourages consistency of error contributions among diseases of the same type (positive or negative), and it penalizes a single severe mistake as strongly as BCE does, while being less sensitive to the accumulation of many small errors. Consequently, $\mathcal{L}_{\text{point}}$ focuses more on the overall error pattern rather than treating each category independently, which is particularly suitable for medical diagnoses where a single critical error should be highlighted, yet multiple moderate errors should also be considered jointly.

A.3.2. GRADIENT BEHAVIOR AND OPTIMIZATION STABILITY

The gradient with respect to $\mu_c$ is:

$$\frac{\partial \mathcal{L}_{\text{point}}}{\partial \mu_c} = \begin{cases} \dfrac{1}{1 + \sum_{c':y_{c'}=0} r_{c'}} \cdot \dfrac{1}{(1 - \mu_c)^2}, & y_c = 0, \\[3mm] -\dfrac{1}{1 + \sum_{c':y_{c'}=1} r_{c'}} \cdot \dfrac{1}{\mu_c^2}, & y_c = 1. \end{cases} \tag{34}$$

The second term amplifies error samples, while the first term normalizes by total error. When a single error dominates, $\sum_{c'} r_{c'} \sim \mathcal{O}(1/\mu_c)$, yielding an overall scaling $\mathcal{O}(1/\mu_c)$, matching BCE. When multiple errors coexist, the first term increases with $\sum_{c'} r_{c'}$, further suppressing individual gradients, preventing noisy sample from dominating. In contrast, BCE lacks such normalization and is more sensitive to isolated noise.

A.3.3. CONVEXITY CONSIDERATION

While each individual term $\log(1 + r_c)$ is convex in its own $\mu_c$, the aggregated form $\log(1 + \sum r_c)$ is not necessarily convex in the vector of all $\mu_c$ when the number of categories per group exceeds two. Nevertheless, the bounded gradients and the practical success of non-convex objectives in deep learning make this a non-critical issue for optimization.

Thus, $\mathcal{L}_{\text{point}}$ provides a balanced and stable objective that emphasizes both severe individual errors and collective error patterns, making it well-suited for fuzzy alignment in multi-label medical image diagnosis.

# B. Details of Methods

## B.1. Details of Semantic Anchor Construction

In Section 3.3, we utilize a GCN to inject pathological dependencies into the semantic anchors. The adjacency matrix $\mathbf{A} \in \mathbb{R}^{C \times C}$ encodes the conditional probability of disease co-occurrence, estimated empirically from the training set statistics. Let $N_i$ denote the total frequency of disease $i$, and $N_{ij}$ denote the frequency of the co-occurrence of diseases $i$ and $j$. To filter out sporadic correlations and noise, we define the edge weights using a thresholded conditional probability:

$$A_{ij} = \mathbb{I}\left(\frac{N_{ij}}{N_i} \geq \eta\right) \cdot \frac{N_{ij}}{N_i}, \tag{35}$$

where $\mathbb{I}(\cdot)$ is the indicator function and $\eta = 0.1$ serves as the noise threshold. Note that this formulation yields an asymmetric matrix, effectively capturing directional dependencies. The input node features $\mathbf{S}^{(0)}$ are initialized using the text embeddings of each disease category name, extracted via the text encoder of a standard pre-trained CLIP (Radford et al., 2021). The semantic propagation is parameterized by a 2-layer GCN with ReLU activations, which projects the linguistic embeddings into the topology-aligned metric space $\mathbb{R}^d$.

## B.2. Details of Training Process

he overall training procedure of FACT is summarized in Algorithm 1. During training, the model parameters are optimized by minimizing the proposed fuzzy objective, which jointly enforces metric alignment and structural regularization. At inference time, the learned fuzzy membership $\mu_c$ corresponding to the $c$-th pathology is directly interpreted as its prediction probability.

---

**Algorithm 1** FACT: Fuzzy Alignment with Comorbidity Topology

---

**Require:** Medical image $x$, disease labels $\mathbf{y} \in \{0,1\}^C$, encoder $\mathcal{F}_{\text{encoder}}$, projection head $\mathcal{F}_{\text{proj}}$, atomic visual codebook $\mathcal{P} = \{p_k\}_{k=1}^K$, initial disease embeddings $\mathbf{S}^{(0)}$, comorbidity graph $\mathbf{A}$.
**Ensure:** Fuzzy membership scores $\boldsymbol{\mu} \in [0,1]^C$.

1: $\boldsymbol{F}_{\text{img}} \leftarrow \mathcal{F}_{\text{encoder}}(x)$
2: $\boldsymbol{Z} \leftarrow \mathcal{F}_{\text{proj}}(\boldsymbol{F}_{\text{img}})$ $\{\boldsymbol{Z} \in \mathbb{R}^{M \times d}\}$

3: **for** $m = 1$ to $M$ **do**
4:     $z_m^q \leftarrow \arg\min_{p_k \in \mathcal{P}} \|z_m - p_k\|_2$
5: **end for**
6: $\mathcal{V} \leftarrow \text{Linear}(\boldsymbol{Z} \oplus \boldsymbol{Z}^q)$ $\{\text{Atomic visual evidence}\}$

7: **for** $l = 0$ to $L - 1$ **do**
8:     $\mathbf{S}^{(l+1)} \leftarrow \text{ReLU}\left(\tilde{\mathbf{D}}^{-\frac{1}{2}} \tilde{\mathbf{A}} \tilde{\mathbf{D}}^{-\frac{1}{2}} \mathbf{S}^{(l)} \mathbf{W}^{(l)}\right)$
9: **end for**
10: $\mathcal{S} \leftarrow \{s_c\}_{c=1}^C$ from $\mathbf{S}^{(L)}$

11: $E \leftarrow Q + \text{Sigmoid}\left(\frac{Q\mathcal{V}^\top}{\sqrt{d}}\right)\mathcal{V}$
12: $\{e_c\}_{c=1}^C \leftarrow E$

13: **for** $c = 1$ to $C$ **do**
14:     $\mu_c \leftarrow \exp\left(-\frac{\|e_c - s_c\|^2}{\tau}\right)$
15: **end for**

16: Compute $\mathcal{L}_{\text{fuzzy}}$ using point-level and set-level fuzzy losses
17: Compute $\mathcal{L}_{\text{commit}}$ for atomic visual space
18: Update model parameters by minimizing $\mathcal{L}_{\text{total}} = \mathcal{L}_{\text{fuzzy}} + \gamma \mathcal{L}_{\text{commit}}$
19: **Return** $\boldsymbol{\mu}$

---

# C. More Details about Datasets and Baselines

## C.1. Datasets

To verify the effectiveness and superiority of our proposed FACT, we use three widely-used MLD datasets to conduct experiments. The introduction of these datasets is given as follows:

- **ODIR-5K** (Li et al., 2021) is an ophthalmic fundus dataset including 5,000 patient fundus images labeled into 8 categories of ocular diseases, including Normal (Norm), Diabetes(Diab), Glaucoma (Glau), Cataract (Catar), AMD, Hypertension(HTN), Myopia(Myop), and Abnormalities (Abn). To illustrate the data imbalance, we visualize the class distribution in Figure 7 (a). Furthermore, the physiological dependencies among diseases are captured in the correlation matrix shown in Figure 8 (a).

- **NIH-Chest** (Wang et al., 2017), a thoracic dataset containing 112,120 frontal chest radiographs from 30,805 patients with 14 pathologies, including Atelectasis (Atelec), Cardiomegaly (Cardio), Effusion (Effus), Infiltration (Infilt), Mass, Nodule (Nod), Pneumonia (Pneum), Pneumothorax (Pneumo), Consolidation (Consol), Edema, Emphysema (Emphy), Fibrosis (Fibro), Pleural Thickening (PleuThick), and Hernia. The disease distribution of this dataset is shown in Figure 7 (b), and the disease correlation matrix is calculated and displayed in Figure 8 (b).

- **RSNA-IHD** (Flanders et al., 2020), a large-scale head CT dataset with 752,802 slices covering 5 hemorrhage subtypes,

including Epidura (Epid), Intraparenchymal (Intrapar), Intraventricular (Intravent), Subarachnoid (Subarach), and Subdural (Subdur). The disease distribution of this dataset is shown in Figure 7 (c), and the disease correlation matrix is calculated and displayed in Figure 8 (c).

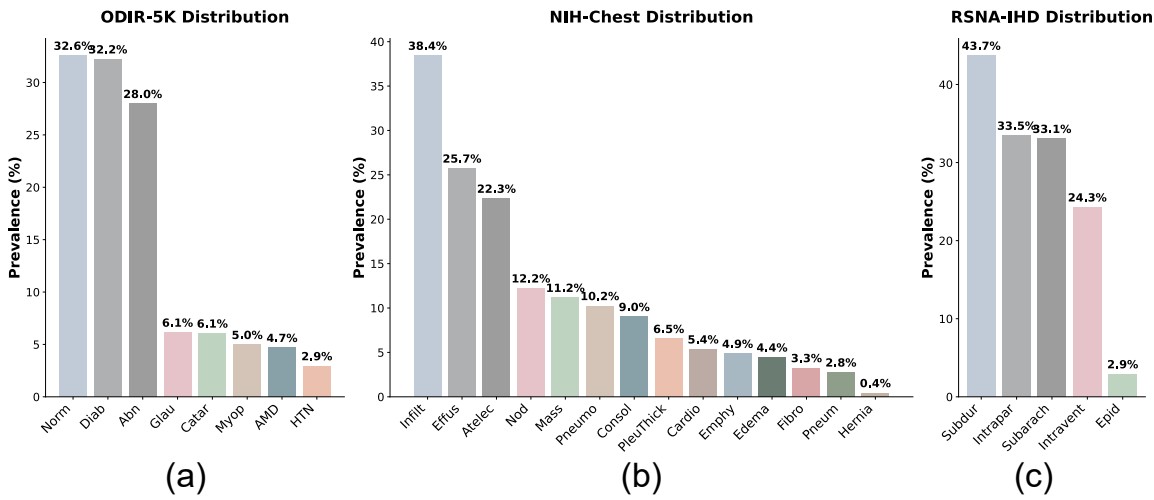

*Figure 7.* Disributions of disease labels.

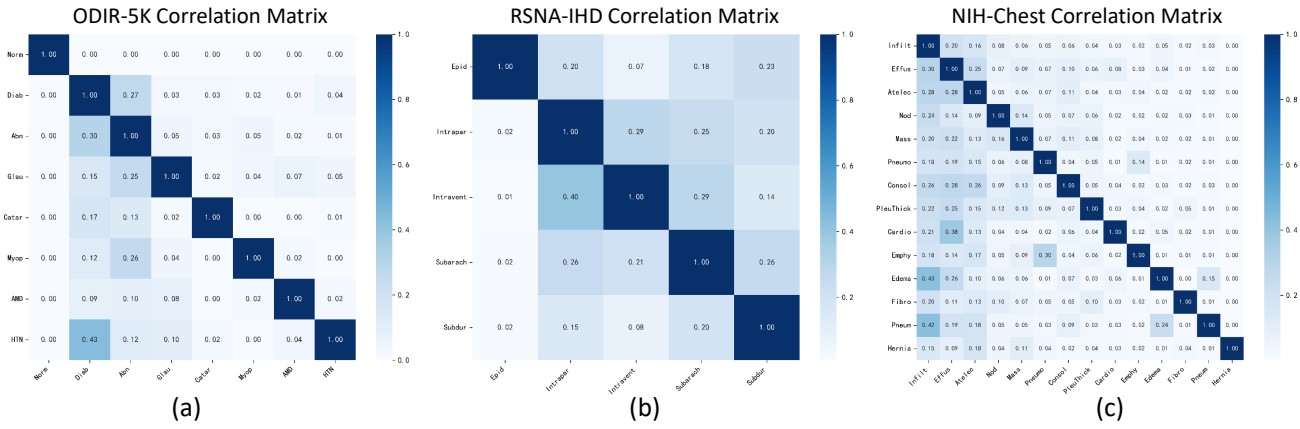

*Figure 8.* Correlation matrices of disease labels.

In real-world medical applications, clinical datasets often suffer from severe class imbalance and noisy annotations, which poses additional challenges for reliable MLD. To further evaluate the effectiveness and robustness of the proposed FACT under long-tailed and noisy conditions, we additionally conduct experiments on CXR-LT (Holste et al., 2025), a large-scale long-tailed and noisy chest X-ray benchmark. CXR-LT contains 377,110 chest X-ray images with 40 disease categories, including Adenopathy, Atelectasis, Azygos Lobe, Calcification of the Aorta, Cardiomegaly, Clavicle Fracture, Consolidation, Edema, Emphysema, Enlarged Cardiomediastinum, Fibrosis, Fissure, Fracture, Granuloma, Hernia, Hydropneumothorax, Infarction, Infiltration, Kyphosis, Lobar Atelectasis, Lung Lesion, Lung Opacity, Mass, Nodule, Normal, Pleural Effusion, Pleural Other, Pleural Thickening, Pneumomediastinum, Pneumonia, Pneumoperitoneum, Pneumothorax, Pulmonary Embolism, Pulmonary Hypertension, Rib Fracture, Round(ed) Atelectasis, Subcutaneous Emphysema, Support Devices, Tortuous Aorta, Tuberculosis. CXR-LT provides two evaluation tasks for multi-label diagnosis. CXR-LT provides two evaluation tasks for multi-label diagnosis:

- **CXR-LT Task 1** is a large-scale noisy long-tailed setting with 40 labels, including 258,871 training images and 78,946 test images. In this task, both training and test labels are automatically extracted from radiology reports and therefore contain annotation noise.

- **CXR-LT Task 2** is a noisy-train clean-test setting with 26 labels, using the same 258,871 training images and 406 test images. In this task, training labels are automatically extracted from radiology reports, while test labels are manually annotated as a gold-standard test set.

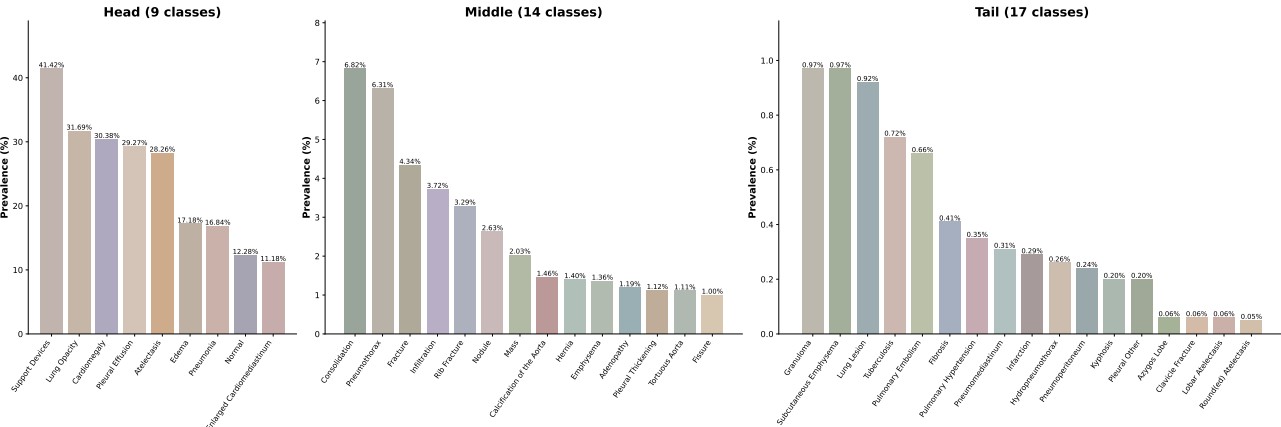

*Figure 9.* Prevalence distribution of disease categories in CXR-LT grouped into Head, Middle, and Tail classes.

Following the long-tailed evaluation protocol, disease categories are grouped by prevalence into Head, Middle, and Tail categories. Head classes have prevalence $\geq 10\%$, Middle classes have prevalence between $1\%$ and $10\%$, and Tail classes have prevalence $< 1\%$. This gives 9 head classes, 14 middle classes, and 17 tail classes, with average prevalence values of 24.82%, 2.7%, and 0.4%, respectively. The detailed per-category prevalence distribution is shown in Figure 9.

## C.2. Baselines

### C.2.1. BASELINE DESCRIPTIONS

To verify the effectiveness and robustness of our proposed FACT, we provide comparison results with 10 baselines that have published code. We introduce each baseline as follows, and the summary of these methods is provided in Table 10.

**Network-centric Approaches:** These methods focus on designing specialized architectures to enhance feature extraction and interaction.

- **TransferNet** (Gour & Khanna, 2021) is an early framework tailored for ophthalmic multi-label diagnosis. It leverages transfer learning with pre-trained backbones (e.g., ResNet) to independently extract features from left and right fundus images, which are subsequently concatenated for joint prediction.

- **TADCL** (Zhang et al., 2023) is a Transformer-based method designed for medical MLD. It constructs a hybrid label representation and proposes a triplet attention mechanism to mitigate sample imbalance and improve generalization to unseen pathologies.

- **CTransCNN** (Wu et al., 2023) adopts a hybrid architecture combining the local locality of CNNs with the global context of Transformers. It introduces a feature interaction module to facilitate bidirectional information exchange between the two streams.

- **LDR** (Huang et al., 2024) proposes a two-stage framework for medical image recognition. It utilizes a CNN for initial feature extraction and a Transformer for semantic decoupling, employing a label reconstruction strategy to alleviate the long-tailed class imbalance.

- **HydraViT** (Öztürk et al., 2025) is an adaptive multi-branch transformer architecture. It integrates a convolutional backbone with a transformer-based context encoder and features a multi-branch output module with learned weighting to capture diverse disease patterns effectively.

**Objective-centric Approaches:** These methods focus on designing robust loss functions or optimization strategies to handle data imbalance and discriminability.

- **ASL** (Ridnik et al., 2021) is a prominent loss function originally proposed for natural images. It introduces asymmetric focusing coefficients to address the severe positive-negative imbalance by dynamically down-weighting easy negative samples.

- **RAL** (Park et al., 2023) extends ASL to further tackle the long-tailed distribution problem. It incorporates Hill loss regularization to enhance model robustness against head-class dominance and label noise.

- **Two-Way** (Kobayashi, 2023) incorporates relative class comparisons into the optimization objective. It constructs a joint discrimination loss that simultaneously optimizes decision boundaries across both class-wise and sample-wise dimensions.

- **SupCon** (Zhang & Wu, 2024) adapts contrastive learning to the supervised multi-label setting. It proposes an efficient multi-label contrastive loss to maximize the separability of features belonging to distinct classes in the embedding space.

- **MultiCo** (Xu et al., 2025) introduces a multi-label voxel-level contrastive loss for medical image analysis. Although originally designed for segmentation, its ability to capture fine-grained semantic differences makes it highly adaptable and effective for MLD tasks.

*Table 10.* Summary of the characteristics of the baseline methods and our proposed FACT framework. We categorize the methods into Network-centric and Objective-centric paradigms. The table details the publication reference, backbone architecture, and computational complexity measured in Parameters (M) and Flops (G). Note that standard ResNet-50 is adopted as the backbone for most methods to ensure a fair comparison, unless the method relies on specific architectures.

| Methods | Type | Ref. | Backbone | Param (M) | Flops (G) |
|---|---|---|---|---|---|
| TransferNet (2021) | Network- centric | BSPC'21 | ResNet50 | 23.52 | 4.13 |
| ASL (2021) | Objective-centric | CVPR'21 | ResNet50 | 23.52 | 4.13 |
| RAL (2023) | Objective-centric | ICCV'23 | ResNet50 | 23.52 | 4.13 |
| TADCL (2023) | Network- centric | MIA'23 | ViT | 61.36 | 8.84 |
| Two-Way (2023) | Objective-centric | CVPR'23 | ResNet50 | 23.52 | 4.13 |
| CTransCNN (2023) | Network- centric | KBS'23 | CNN+ViT | 33.99 | 10.57 |
| LDR (2024) | Network- centric | ACMMM'24 | ResNet50 | 25.51 | 8.42 |
| SupCon (2024) | Objective-centric | AAAI'24 | ResNet50 | 23.52 | 4.13 |
| MultiCo (2025) | Objective-centric | arxiv'25 | ResNet50 | 23.52 | 4.13 |
| HydraViT (2025) | Network- centric | BSPC'25 | ResNet50 | 39.22 | 16.58 |
| FACT (Ours) | – | – | ResNet50 | 25.87 | 4.21 |

#### C.2.2. COMPUTATIONAL EFFICIENCY COMPARISON

To assess the efficiency of our proposed framework, we benchmark the model complexity using the RSNA dataset (5 diseases). Compared to the standard ResNet-50 backbone used in baseline methods, FACT introduces only a marginal overhead, with an increase of 2.35M in parameters and 0.08 GFlops in computational cost. These additional costs primarily stem from the learnable semantic interface operator (defined in Eq. 1) and the 2-layer Graph Convolutional Network (GCN) for comorbidity topology fusion (defined in Eq. 7), both of which are lightweight modules. In stark contrast to network-centric approaches such as TADCL (61.36M Params / 8.84 GFlops), CTransCNN (33.99M Params / 10.57 GFlops), and HydraViT (39.22M Params / 16.58 GFlops), FACT maintains a significantly more compact architecture. This confirms that our performance gains are derived from the effective metric-based fuzzy alignment paradigm rather than simply scaling up model capacity, making FACT highly suitable for resource-constrained clinical deployment.

## D. More Results about Performance

To provide a comprehensive evaluation, we detail the per-class AUC performance in Tables 11, 12, and 13. Note that the average values here are calculated by averaging individual class scores, yielding results distinct from the global flattening approach used in Table 1. This fine-grained analysis is crucial for diagnostic reliability: it explicitly demonstrates that our

*Table 11.* The quantitative comparisons with SOTA MLD methods on the ODIR-5K dataset. The optimal result is represented in bold, the second-best result is underlined.

| Dieases | Methods | | | | | | | | | | | |
|---|---|---|---|---|---|---|---|---|---|---|---|---|
| | TransferNet (2021) | ASL (2021) | RAL (2023) | TADCL (2023) | Two-Way (2023) | CTransCNN (2023) | LDR (2024) | SupCon (2024) | MultiCo (2025) | HydraViT (2025) | Ours | △ |
| Norm | 80.45 | 81.87 | 82.59 | 80.29 | 82.73 | 81.98 | 83.13 | 82.63 | 82.92 | 83.99 | **86.48** | +2.49 |
| Diab | 85.38 | 80.70 | 80.30 | 80.43 | 81.60 | 82.36 | 83.28 | 80.40 | 85.34 | 85.14 | **85.80** | -0.58 |
| Abn | 73.26 | 77.46 | **81.93** | 78.33 | 81.78 | 78.92 | 78.02 | 79.85 | 79.57 | 80.46 | 81.47 | +0.42 |
| Glau | 93.44 | 89.14 | 90.81 | 90.59 | 92.29 | 90.12 | 90.41 | 94.98 | 93.82 | 87.68 | **95.11** | +0.13 |
| Catar | 98.38 | 97.82 | 97.90 | 97.51 | 95.65 | 96.11 | 97.09 | 97.79 | 97.41 | 97.63 | **98.42** | +0.04 |
| Myop | 98.47 | 99.52 | 98.63 | 99.59 | **99.82** | 99.49 | 99.13 | 99.76 | 99.76 | 99.27 | 99.77 | -0.05 |
| AMD | 88.27 | 90.27 | 81.90 | 89.40 | 90.29 | 89.61 | 88.48 | 92.03 | 90.26 | 86.28 | **95.29** | +3.26 |
| HTN | 75.36 | 68.89 | 69.85 | 68.24 | 68.66 | 69.14 | 81.56 | 84.21 | 70.85 | 79.49 | **86.62** | +2.41 |
| **Average** | 86.63 | 85.71 | 85.48 | 85.55 | 86.61 | 85.97 | 87.64 | 88.96 | 87.49 | 87.50 | **91.12** | +2.16 |

*"△" represents the difference in metrics between our method and the second-best method.*

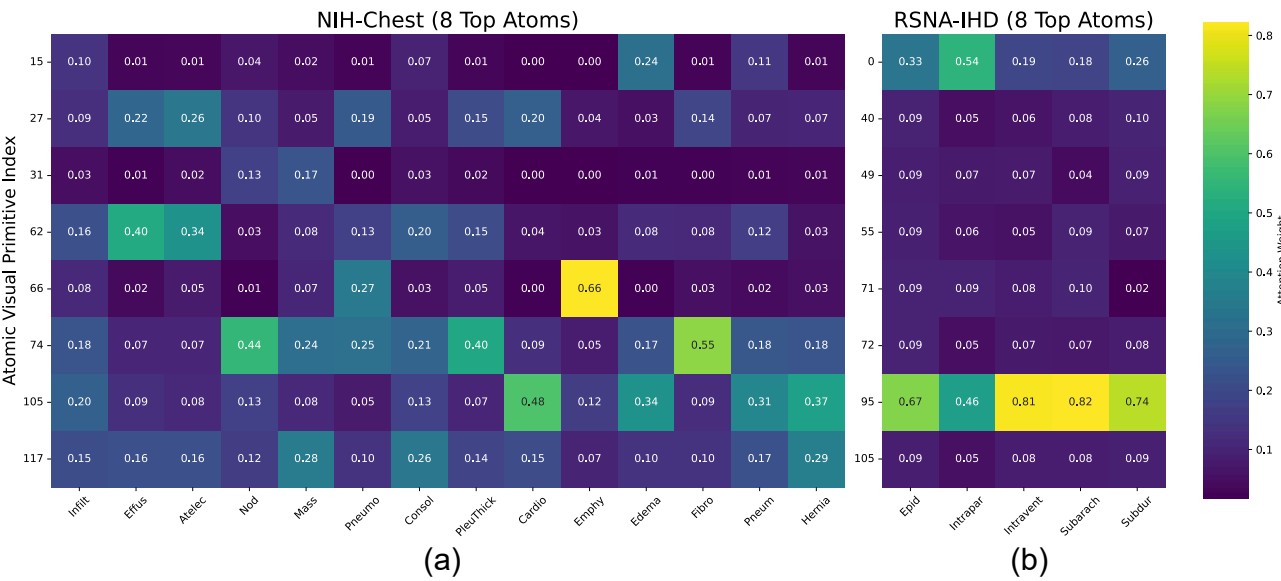

*Figure 10.* Visualization of the activation affinity between disease categories and visual atoms on the NIH-Chest (a) and RSNA-IHD (b).

high overall performance is not driven solely by dominant classes, but stems from robust identification capabilities across every specific disease category.

Table 11 presents the quantitative comparison on the ODIR-5K dataset where FACT achieves a superior mean AUC of 91.10%. Our framework outperforms the second-best method SupCon by a significant margin of 2.14%. Specifically, FACT demonstrates remarkable robustness in diagnosing complex and visually ambiguous pathologies. In the AMD and Hypertension categories, our method surpasses the nearest competitors by 3.26% and 2.41% respectively. Furthermore, the performance in the Normal category is improved by 2.49% compared to HydraViT, indicating a reduction in false positives. While baselines like RAL show marginal advantages in structural diseases such as Cataract, FACT maintains the highest consistency and overall diagnostic accuracy across the diverse ocular conditions.

For the NIH-Chest dataset (Table 12), our framework secures the highest average AUC of 82.60%. While baselines like HydraViT and CTransCNN show competitive results in structural abnormalities like Mass or Nodule, FACT exhibits superior capability in diagnosing diffuse and overlapping textual patterns. Most notably, we observe a dominant lead in the Pneumonia category (+3.00% vs CTransCNN) and Emphysema (+0.93% vs ASL). This confirms that the proposed fuzzy alignment mechanism effectively disentangles subtle visual evidence in challenging thoracic pathologies.

*Table 12.* The quantitative comparisons with SOTA MLD methods on the NIH-Chest dataset. The optimal result is represented in bold, the second-best result is underlined.

| Dieases | Methods | | | | | | | | | | | |
|---------|---------|-----|-----|-------|---------|----------|-----|--------|---------|----------|------|-----|
| | TransferNet (2021) | ASL (2021) | RAL (2023) | TADCL (2023) | Two-Way (2023) | CTransCNN (2023) | LDR (2024) | SupCon (2024) | MultiCo (2025) | HydraViT (2025) | Ours | △ |
| Infilt | 71.35 | 69.48 | 68.96 | 69.90 | 71.14 | 69.54 | 71.66 | 71.44 | 70.62 | 70.50 | **72.32** | +0.66 |
| Effus | 81.72 | 79.68 | 76.70 | 79.63 | 82.24 | 83.22 | 83.70 | 83.27 | 83.26 | 82.28 | **84.41** | +0.71 |
| Atelec | 78.58 | 76.20 | 75.17 | 75.21 | 77.57 | 78.62 | 79.31 | 78.70 | 78.37 | 78.75 | **79.69** | +0.38 |
| Nod | 76.93 | 76.18 | 75.97 | 75.13 | 77.89 | 77.34 | 78.35 | 78.31 | 76.12 | **78.90** | 78.66 | -0.24 |
| Mass | 81.36 | 81.22 | 81.65 | 80.62 | 81.95 | 83.46 | 83.08 | 83.36 | 83.29 | 83.99 | **84.09** | +0.10 |
| Pneumo | 86.15 | 87.16 | 86.00 | 82.06 | 85.42 | 85.43 | 86.60 | 86.76 | 87.04 | **87.61** | 87.18 | -0.43 |
| Consol | 69.99 | 66.88 | 62.39 | 66.11 | 69.08 | **72.90** | 72.34 | 72.51 | 72.67 | 70.93 | 72.85 | -0.05 |
| PleuThick | 74.38 | 73.85 | 71.54 | 71.38 | 73.93 | 78.32 | 77.40 | 74.65 | 73.81 | 75.26 | **78.07** | +0.67 |
| Cardio | 89.60 | 88.57 | 86.29 | 86.63 | **90.51** | 84.39 | 87.00 | 88.71 | 89.73 | 89.84 | 89.68 | -0.83 |
| Emphy | 90.23 | 91.25 | 89.86 | 90.99 | 90.52 | 88.10 | 89.86 | 90.64 | 90.75 | 90.51 | **92.18** | +0.93 |
| Edema | 86.11 | 83.77 | 81.21 | 82.23 | 86.54 | 85.79 | 84.78 | 85.10 | 86.57 | **87.10** | 86.78 | -0.32 |
| Fibro | 81.56 | 79.97 | 80.16 | 79.24 | 80.54 | 78.12 | 74.98 | 80.39 | 82.14 | 79.70 | **81.58** | +0.02 |
| Pneum | 67.04 | 65.11 | 58.76 | 64.55 | 68.73 | 71.62 | 71.33 | 70.48 | 68.52 | 65.74 | **74.62** | +3.00 |
| Hernia | 82.88 | 94.77 | 95.02 | 90.28 | 86.22 | 84.78 | 91.31 | 93.53 | 87.27 | 89.59 | **94.94** | +0.17 |
| **Average** | 79.85 | 79.58 | 77.83 | 78.14 | 80.16 | 80.12 | 80.84 | 81.28 | 80.73 | 80.76 | **82.65** | +1.37 |

*"△" represents the difference in metrics between our method and the second-best method.*

*Table 13.* The quantitative comparisons with SOTA MLD methods on the RSNA-IHD dataset. The optimal result is represented in bold, the second-best result is underlined.

| Dieases | Methods | | | | | | | | | | | |
|---------|---------|-----|-----|-------|---------|----------|-----|--------|---------|----------|------|-----|
| | TransferNet (2021) | ASL (2021) | RAL (2023) | TADCL (2023) | Two-Way (2023) | CTransCNN (2023) | LDR (2024) | SupCon (2024) | MultiCo (2025) | HydraViT (2025) | Ours | △ |
| Epid | 94.66 | 95.60 | 94.63 | 96.93 | 91.39 | 94.77 | 93.32 | 91.01 | 94.57 | 90.92 | **97.23** | +0.30 |
| Intrapar | 95.16 | 94.52 | 94.83 | 95.74 | 93.65 | 95.74 | 94.56 | 95.47 | 95.59 | 95.64 | **96.07** | +0.33 |
| Intravent | 96.82 | 96.65 | 97.51 | 97.71 | 97.39 | 97.01 | 97.33 | 97.72 | 96.79 | 97.01 | **97.73** | +0.01 |
| Subarach | 92.69 | 91.85 | 90.72 | 93.30 | 88.08 | 91.51 | 90.28 | 91.19 | 92.36 | 91.33 | **93.92** | +0.62 |
| Subdur | 95.04 | 93.32 | 93.44 | 95.74 | 91.52 | 93.48 | 91.88 | 93.15 | 94.83 | 93.81 | **96.32** | +0.58 |
| **Average** | 94.87 | 94.39 | 94.23 | 95.89 | 92.41 | 94.50 | 93.47 | 93.71 | 94.83 | 93.74 | **96.25** | +0.35 |

*"△" represents the difference in metrics between our method and the second-best method.*

On the RSNA-IHD dataset detailed in Table 13, FACT maintains consistent superiority across hemorrhage subtypes and secures the highest scores in four out of five categories. We observe a distinct advantage of 0.62% in Subarachnoid hemorrhage detection, a task often complicated by the subtle spread of blood along the brain surface. The sustained performance across diverse modalities including fundus photography, X-rays, and CT scans confirms the robust generalization capability of our comorbidity-aware fuzzy alignment paradigm.

For the CXR-LT dataset (Table 14), FACT achieves strong per-disease performance under severe long-tailed and noisy annotations. On head categories, FACT remains competitive with the strongest baselines, although HydraViT obtains slightly better results on several frequent diseases such as Cardiomegaly, Edema, and Pleural Effusion. In contrast, FACT shows clearer advantages on middle and tail categories, where the number of positive samples is much smaller and labels are noisier. For middle classes, FACT obtains notable improvements on Fissure (+8.61%), Fracture (+3.55%), Rib Fracture (+3.26%), Mass (+2.21%), and Nodule (+2.00%). For tail classes, FACT achieves larger gains on several rare diseases, including Azygos Lobe (+12.11%), Clavicle Fracture (+11.56%), Pneumoperitoneum (+5.58%), Infarction (+4.77%), and Pneumomediastinum (+3.35%). These results further support the robustness of FACT under long-tailed and noisy clinical data settings.

*Table 14.* The quantitative comparisons with SOTA MLD methods on the long-tail and noisy CXR-LT Task 1. The optimal result is represented in bold, the second-best result is underlined.

| Diseases | Prevalence | Group | MultiCo(2025) | LDR(2024) | HydraViT(2025) | Ours | △ |
|---|---|---|---|---|---|---|---|
| Adenopathy | 1.19% | Middle | 68.50 | 70.69 | 76.28 | **78.55** | +2.27 |
| Atelectasis | 28.26% | Head | 79.42 | 80.05 | 80.11 | **80.19** | +0.08 |
| Azygos Lobe | 0.06% | Tail | 69.77 | 73.28 | 69.72 | **85.39** | +12.11 |
| Calcification of the Aorta | 1.46% | Middle | 76.90 | 85.88 | 81.74 | **87.95** | +2.07 |
| Cardiomegaly | 30.38% | Head | 79.04 | 79.45 | **79.52** | 78.58 | -0.94 |
| Clavicle Fracture | 0.06% | Tail | 58.55 | 62.96 | 64.67 | **76.23** | 11.56 |
| Consolidation | 6.82% | Middle | 74.90 | 76.62 | **76.98** | 76.60 | -0.38 |
| Edema | 17.18% | Head | 82.82 | 83.39 | **84.15** | 83.40 | -0.75 |
| Emphysema | 1.36% | Middle | 82.00 | 86.98 | **88.60** | 88.19 | 0.41 |
| Enlarged Cardiomediastinum | 11.18% | Head | 55.34 | 55.49 | **59.45** | 58.61 | -0.84 |
| Fibrosis | 0.41% | Tail | 76.40 | 85.06 | 87.89 | **89.21** | +1.32 |
| Fissure | 1.00% | Middle | 59.03 | 67.64 | 68.02 | **76.63** | +8.61 |
| Fracture | 4.34% | Middle | 64.02 | 68.41 | 70.50 | **74.05** | +3.55 |
| Granuloma | 0.97% | Tail | 70.36 | 73.46 | 73.82 | **76.16** | +2.34 |
| Hernia | 1.40% | Middle | 74.88 | **85.77** | 84.76 | 85.57 | -0.20 |
| Hydropneumothorax | 0.26% | Tail | 74.78 | 90.67 | 93.14 | **93.82** | +0.68 |
| Infarction | 0.29% | Tail | 49.14 | 51.67 | 56.27 | **61.04** | +4.77 |
| Infiltration | 3.72% | Middle | 54.65 | 56.80 | 57.71 | **58.10** | +0.39 |
| Kyphosis | 0.20% | Tail | 78.13 | 87.99 | 86.07 | **90.19** | +2.20 |
| Lobar Atelectasis | 0.06% | Tail | 48.99 | 82.26 | 78.35 | **85.03** | +2.77 |
| Lung Lesion | 0.92% | Tail | 65.33 | 69.70 | 73.82 | **76.36** | +2.54 |
| Lung Opacity | 31.69% | Head | 74.07 | 74.83 | 75.64 | **75.77** | +0.13 |
| Mass | 2.03% | Middle | 70.21 | 73.62 | 75.34 | **77.45** | +2.21 |
| Nodule | 2.63% | Middle | 69.90 | 72.41 | 72.91 | **74.91** | +2.00 |
| Normal | 12.28% | Head | 79.27 | 78.98 | **79.58** | 79.42 | -0.16 |
| Pleural Effusion | 29.27% | Head | 89.42 | 89.89 | **90.42** | 89.92 | -0.50 |
| Pleural Other | 0.20% | Tail | 64.56 | 81.37 | 82.13 | **83.49** | +1.36 |
| Pleural Thickening | 1.12% | Middle | 71.28 | 77.81 | 79.55 | **83.25** | +0.70 |
| Pneumomediastinum | 0.31% | Tail | 56.93 | 73.61 | 80.06 | **83.41** | +3.35 |
| Pneumonia | 16.84% | Head | 60.35 | 61.92 | **62.68** | 62.49 | -0.19 |
| Pneumoperitoneum | 0.24% | Tail | 61.81 | 71.14 | 74.33 | **79.91** | +5.58 |
| Pneumothorax | 6.31% | Middle | 78.86 | 80.56 | 82.67 | **82.70** | +0.03 |
| Pulmonary Embolism | 0.66% | Tail | 49.56 | 51.52 | 58.94 | **59.37** | +0.43 |
| Pulmonary Hypertension | 0.35% | Tail | 58.81 | 75.02 | 76.71 | **77.94** | +1.23 |
| Rib Fracture | 3.29% | Middle | 62.23 | 67.68 | 70.88 | **74.14** | +3.26 |
| Round(ed) Atelectasis | 0.05% | Tail | 57.21 | 92.11 | **92.59** | 90.18 | -2.41 |
| Subcutaneous Emphysema | 0.97% | Tail | 90.49 | 94.16 | 95.59 | **96.08** | +0.49 |
| Support Devices | 41.42% | Head | 91.79 | 91.75 | 91.85 | **92.53** | +0.68 |
| Tortuous Aorta | 1.11% | Middle | 73.47 | 75.63 | 78.35 | **79.68** | +1.33 |
| Tuberculosis | 0.72% | Tail | 69.28 | 71.47 | 76.03 | **76.60** | +0.57 |

*"△" represents the difference in metrics between our method and the second-best method.*

*Table 15.* Analysis of VPS utilization and quantization error across datasets. Ours$^\dagger$ denotes FACT with dead-code reset during VQ training. "Used" denotes the number of activated visual atomic primitives, "Uniq./Img." denotes the average number of unique primitives used per image, and "Quant. Err." denotes the average quantization error.

| Method | ODIR-5K | | | NIH-Chest | | | RSNA-IHD | | | CXR-LT | | |
|---|---|---|---|---|---|---|---|---|---|---|---|---|
| | Used | Uniq./Img. | Quant. Err. | Used | Uniq./Img. | Quant. Err. | Used | Uniq./Img. | Quant. Err. | Used | Uniq./Img. | Quant. Err. |
| Ours | 7 | 4.63 | $8.47 \times 10^{-4}$ | 10 | 5.38 | $6.67 \times 10^{-4}$ | 9 | 4.34 | $4.06 \times 10^{-4}$ | 11 | 8.09 | $7.75 \times 10^{-4}$ |
| Ours$^\dagger$ | 93 | 30.50 | $\mathbf{2.46 \times 10^{-4}}$ | 56 | 16.06 | $\mathbf{5.62 \times 10^{-4}}$ | 81 | 17.88 | $\mathbf{3.25 \times 10^{-4}}$ | 103 | 20.88 | $\mathbf{6.49 \times 10^{-4}}$ |

*Table 16.* Ablation study on dead-code reset strategy.

| Method | ODIR-5K | | | | NIH-Chest | | | | RSNA-IHD | | | | CXR-LT | | | |
|---|---|---|---|---|---|---|---|---|---|---|---|---|---|---|---|---|
| | mAP↑ | F1↑ | AUC↑ | Avg.↑ | mAP↑ | F1↑ | AUC↑ | Avg.↑ | mAP↑ | F1↑ | AUC↑ | Avg.↑ | mAP↑ | F1↑ | AUC↑ | Avg.↑ |
| Ours | 72.61 | **90.91** | **92.97** | 85.50 | **58.95** | **90.44** | **87.81** | **79.07** | 89.72 | **92.97** | **96.90** | **93.20** | 57.54 | **94.79** | 93.08 | 81.80 |
| Ours$^\dagger$ | **74.32** | 90.39 | 92.95 | **85.89** | 58.39 | 89.76 | 87.53 | 78.56 | **90.08** | 92.30 | 96.65 | 93.01 | **57.75** | 94.77 | **93.14** | **81.89** |

## E. More Analysis of Atomic Visual Space

To qualitatively verify whether the learned Atomic Visual Space successfully captures the intrinsic polysemy of pathological signs, we visualize the affinity matrix between disease semantic anchors and the top-activated visual atoms in Figure 10.

As shown in Figure 10 (a), the heatmap reveals a dense pattern of shared activation, confirming that singular visual atoms often serve as compositional evidence for multiple distinct pathologies. A prime example is Atom 74, which exhibits high affinity for Nodule (0.44), Pleural Thickening (0.40), and Fibrosis (0.55). This shared activation pattern is clinically consistent, as these conditions all manifest as localized tissue densification and scarring. Similarly, Atom 62 bridges Effusion (0.40) and Atelectasis (0.34), reflecting their frequent physiological concurrence. This proves that FACT does not memorize rigid one-to-one mappings but instead learns to identify fundamental atomic visual evidence shared across comorbidity clusters.

As shown in Figure 10 (b), the distribution is more concentrated in the intracranial hemorrhage task, reflecting the high morphological similarity among hemorrhage subtypes. We identify Atom 95 and Atom 0 as representative visual atoms. Specifically, Atom 95 dominates the evidence spectrum, showing strong metric alignment across Intraventricular (0.81), Subarachnoid (0.82), and Subdural (0.74) hemorrhages. This suggests that Atom 95 encodes the fundamental visual pattern of hyperdense blood common to all subtypes, which demonstrate the effectiveness of the Atomic Visual Space of our FACT.

We further analyze the utilization of the Visual Primitive Space (VPS). Since VQ training may leave many primitives inactive, we adopt a simple dead-code reset strategy to reinitialize rarely used primitives during training. As shown in Table 15, dead-code reset substantially increases VPS usage and the number of unique primitives used per image, while consistently reducing the average quantization error across datasets. Table 16 further shows that better VPS utilization brings slight gains on ODIR-5K and CXR-LT and maintains comparable performance on NIH-Chest and RSNA-IHD. These results support the effectiveness of VPS and indicate that improving the utilization of visual atomic primitives can further benefit FACT.

## F. More Results about Ablation Study

We further analyze the sensitivity of FACT to different encoder compression rates. By varying the compression rate from $8\times$ to $16\times$ and $32\times$, the model constructs atomic visual evidence at different spatial granularities. As reported in Table 17, FACT maintains stable performance across all four datasets, indicating that the proposed framework is not overly sensitive to the encoder compression rate.

On ODIR-5K, the $8\times$ setting achieves the best mAP and Avg., while the $32\times$ setting obtains the best F1, and all three settings remain within a narrow performance range. This suggests that ophthalmic disease diagnosis benefits from relatively fine-grained visual evidence, but FACT remains robust when the visual representation is more compressed. On NIH-Chest, the $32\times$ setting performs best across all metrics, indicating that a more compact representation may be sufficient for capturing global thoracic patterns in chest radiographs. On RSNA-IHD, the $16\times$ setting achieves the best mAP, AUC, and Avg., while $32\times$ yields the best F1, showing that moderate compression provides a favorable trade-off for hemorrhage-related CT diagnosis. On CXR-LT, the performance differences among different compression rates are also small, with $16\times$ slightly outperforming others. Overall, these results demonstrate that FACT consistently performs well under different visual granularities, supporting the robustness of atomic visual evidence construction.

*Table 17.* Ablation study on compression rates.

| Comp. Rate | ODIR-5K | | | | NIH-Chest | | | | RSNA-IHD | | | | CXR-LT | | | |
|---|---|---|---|---|---|---|---|---|---|---|---|---|---|---|---|---|
| | mAP↑ | F1↑ | AUC↑ | Avg.↑ | mAP↑ | F1↑ | AUC↑ | Avg.↑ | mAP↑ | F1↑ | AUC↑ | Avg.↑ | mAP↑ | F1↑ | AUC↑ | Avg.↑ |
| $8\times$ | **74.09** | 90.62 | 92.94 | **85.88** | 56.70 | 89.87 | 87.24 | 77.94 | 90.34 | 92.27 | 96.83 | 93.15 | 57.86 | 94.67 | 93.17 | 81.90 |
| $16\times$ | 73.42 | 90.09 | **93.06** | 85.52 | 56.05 | 89.80 | 87.33 | 77.73 | **90.78** | 92.37 | **97.03** | **93.39** | **58.15** | **94.85** | **93.23** | **82.08** |
| $32\times$ (default) | 72.61 | **90.91** | 92.97 | 85.50 | **58.95** | **90.44** | **87.81** | **79.07** | 89.72 | **92.97** | 96.90 | 93.20 | 57.54 | 94.79 | 93.08 | 81.80 |

To intuitively understand the contribution of each module, we visualize the prediction confidence scores for challenging samples in Figure 11. Comparing the full FACT framework with its ablated variants reveals distinct failure modes in the baseline methods.

In the second column (NIH-Chest), the patient presents with multiple co-occurring pathologies including Edema, Nodule, Infiltration, and Effusion. The variant without the Comorbidity Topology Graph (w/o CTG) fails to detect Infiltration

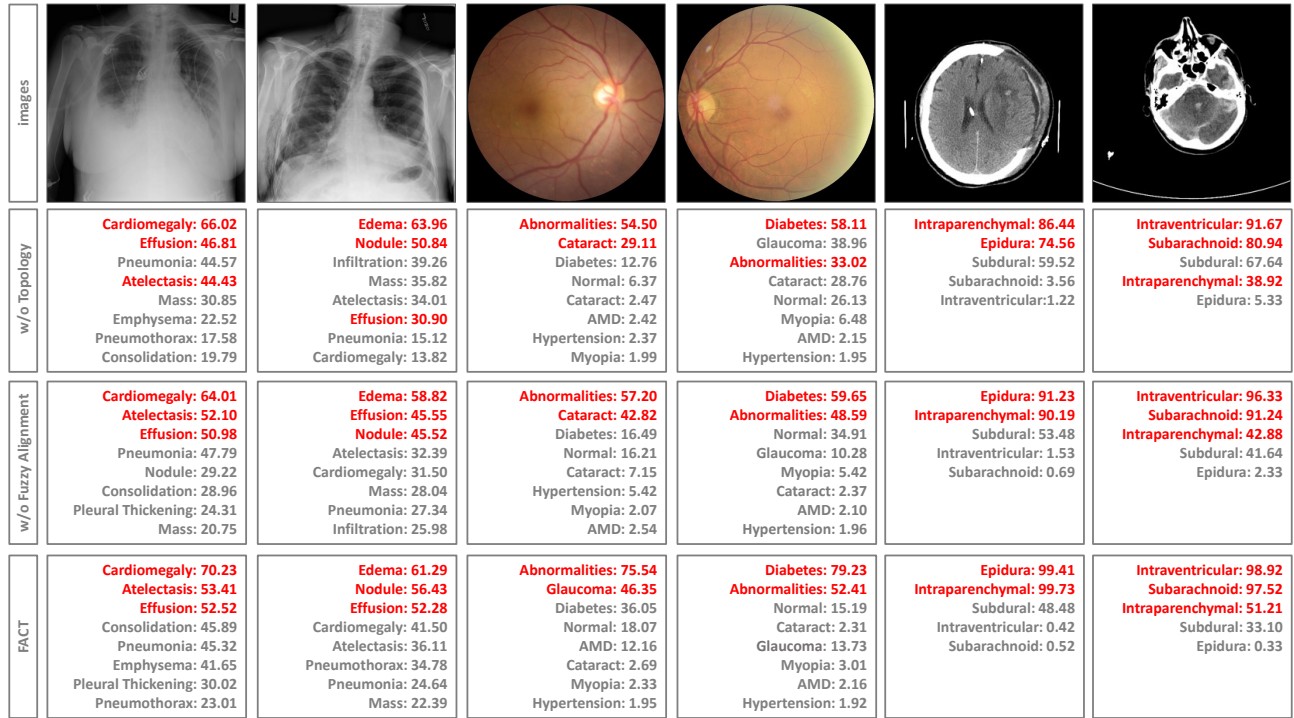

*Figure 11.* Qualitative ablation results on NIH-Chest and ODIR-5K samples. We compare the predicted membership scores of the full FACT model against variants without the Comorbidity Topology and without the Fuzzy Alignment. Ground truth positive diseases are marked in red.

and Effusion. This failure occurs because the model treats diseases independently, ignoring the strong clinical correlation where Edema often accompanies Effusion. Similarly, the variant without Fuzzy Alignment Loss (w/o FAL) produces lower confidence scores for valid labels, indicating a struggle to align ambiguous visual features with semantic concepts. In contrast, the complete FACT model successfully identifies all four ground-truth diseases with high confidence. This comparison qualitatively confirms that the topological prior is essential for reasoning about comorbidities, while the fuzzy alignment mechanism significantly enhances the model's certainty and discriminative power.

## G. Parametric Analysis

To study the impact of different hyperparameter settings on performance, we perform sensitivity analysis on RSNA-IHD for six key hyperparameters, including the network dimension $d$, primitive number $K$ in Atomic Visual Space, weight $\alpha$ of set-level fuzzy loss, balance weight $\beta$ of total loss, weight $\gamma$ of commitment loss, and fuzzy temperature $tau$. The results are illustrated in Figure 12.

**Structural Dimensions** ($d, K$). The model demonstrates robustness to the network dimension $d$, which is shown in Figure 12(a). Performance remains stable across the range $[128, 768]$, with the optimal performance at $d = 512$, where the fluctuation between the highest and lowest scores is contained within a narrow margin of less than 1%. In contrast, the selection of the Atomic Visual Space $K$ exerts a tangible influence on fine-grained recognition, reflected in the mAP metric, as shown in Figure 12(b). We observe a general upward trend as $K$ increases, peaking at 89.16% when $K = 384$ versus a minimum of 87.45% at $K = 32$. This positive correlation suggests that a larger Atomic Visual Space facilitates the learning of a more diverse set of atomic visual evidence, enabling the model to capture subtle variations in pathological signs.

**Loss Balancing Hyperparameters** ($\alpha, \beta, \gamma$). The weight of the set-level fuzzy loss $\alpha$ shows a monotonic improvement trend, reaching saturation at $\alpha = 0.7$, as shown in Figure 12(c). It indicates that explicit structural supervision is crucial for MLD. For the commitment loss weight $\beta$, we observe a concave performance trajectory, where metric peaks at $\beta = 0.5$ and degrades at both extremes ($\beta = 0.1$ and $\beta = 1.0$), as shown in Figure 12(d). This behavior implies a trade-off mechanism that a lower $\beta$ fails to stabilize the learning atomic visual evidence, while a higher $\beta$will overly constrains the encoder,

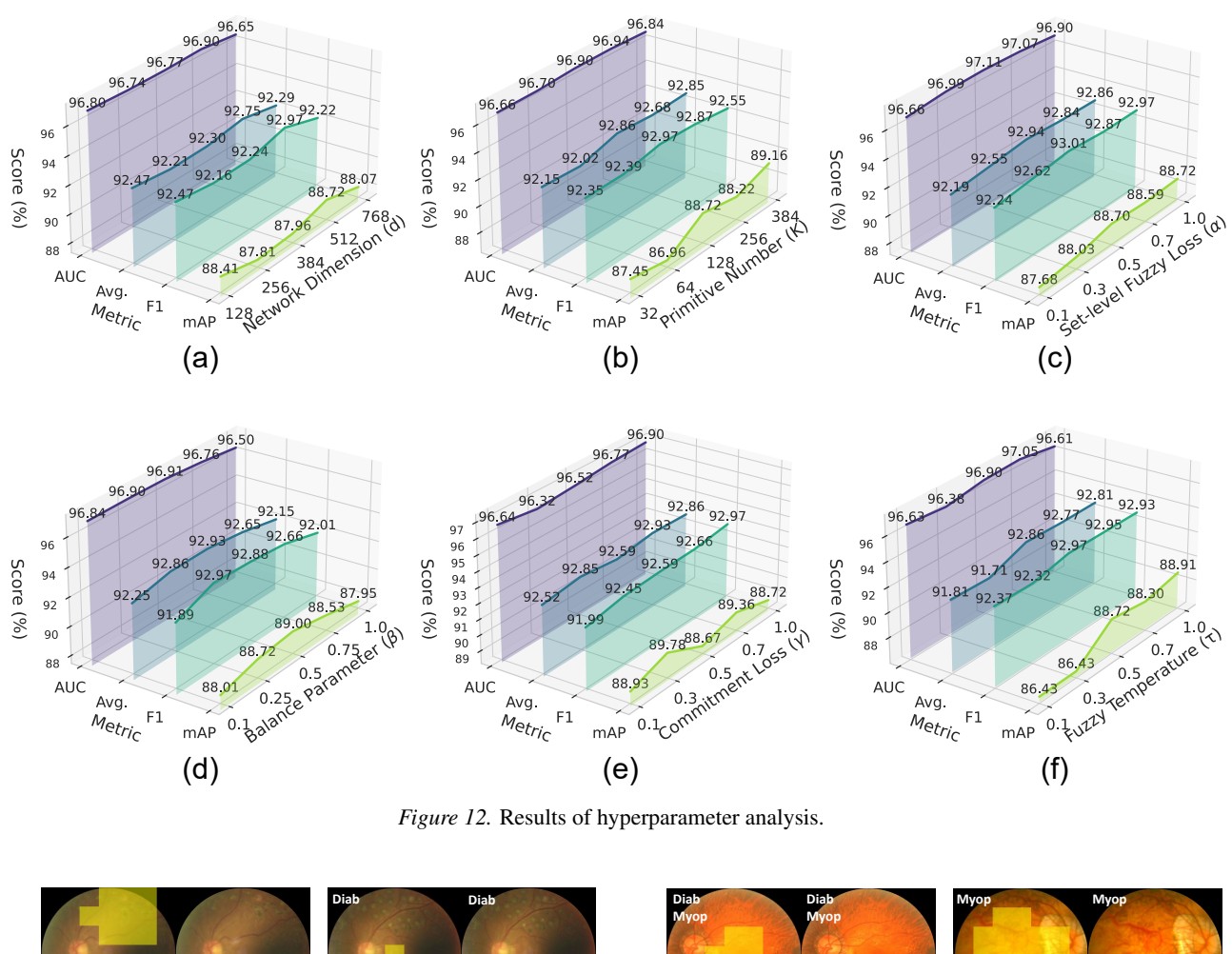

Figure 12. Results of hyperparameter analysis.

Figure 13. Spatial visualization of Atomic Visual Evidence. We back-project the top-activated visual atoms onto the original fundus images to visualize their receptive fields, which are highlighted in yellow.

limiting representation flexibility. Regarding the commitment loss weight $\gamma$, the model is largely insensitive, as shown in 12(e). The mAP exhibits minor fluctuations, global metrics like AUC and Average Score remain highly stable with variations below 0.3%.

**Gaussian Temperature Sensitivity ($\tau$).** As shown in 12(f), the temperature parameter $\tau$ plays a decisive role in the alignment mechanism. We observe a significant performance drop at lower temperatures ($\tau \in \{0.1, 0.3\}$). Theoretically, $\tau$ controls the bandwidth of the Gaussian kernel, where a vanishing $\tau$ renders the membership function hypersensitive to metric deviations. This causes the alignment scores to saturate rapidly towards 0 or 1, leading to vanishing gradients during training and extreme, uncalibrated confidence scores during inference. The optimal stability is achieved at $\tau = 0.5$, which provides an appropriate tolerance radius for the fuzzy alignment on the semantic manifold.

## H. Visualization of Atomic Visual Evidence

To further validate that our Atomic Visual Space captures semantically meaningful primitives rather than abstract noise, we visualize the spatial attention of specific atoms in Figure 13. This visualization result qualitative analysis demonstrates how the model constructs diagnoses compositionally.

As shown in Fig. 13(a), Atom 42 functions as a specialized detector for diabetic retinopathy signs. It consistently activates regions containing hard exudates and microaneurysms. Crucially, this activation is context-invariant. Whether the patient is diagnosed with Diabetes alone or presents with additional abnormalities, Atom 42 reliably identifies the same underlying pathological evidence. This proves that FACT successfully decomposes complex disease labels into their constituent visual signs and treats distinct lesions as the fundamental atomic units of diagnosis.

In Fig. 13(b), Atom 54 demonstrates the model's ability to leverage anatomical landmarks. This atom focuses on the optic disc and the peripapillary region which are critical areas for diagnosing high myopia features such as tessellated fundus or peripapillary atrophy. The model dynamically recruits this atom across different patient profiles by activating it for both pure Myopia cases and complex Diabetes-Myopia comorbidities. This behavior exemplifies the concept of visual polysemy where the visual primitive representing optic disc anomalies serves as a shared resource that supports any diagnostic hypothesis involving myopic degeneration independent of other co-occurring conditions.

