# OpenReview forum: "FACT: Fuzzy Alignment with Comorbidity Topology for Reliable Multi-Label Medical Image Diagnosis"
_ICML.cc/2026/Conference — ICML 2026 regular_

### Official Review · Reviewer_zL8X · 2026-03-10

**Soundness:** 2
**Presentation:** 3
**Significance:** 3
**Originality:** 2
**Overall Recommendation:** 4
**Confidence:** 3

**Summary:**

This paper studies multi-label medical image diagnosis under comorbidity. The main idea is to replace rigid independent classification with a fuzzy alignment framework between disease-specific visual evidence and disease semantic anchors. It has three parts: an atomic visual space built by vector quantization, semantic anchors refined by a comorbidity graph, and a Gaussian membership function with a fuzzy loss for alignment. The paper evaluates the method on three public benchmarks and reports consistent gains over others.

**Compliance With Llm Reviewing Policy:**

Affirmed.

**Final Justification:**

We sincerely thank the authors for their thoughtful and detailed response. Our concerns have been adequately addressed, and we therefore maintain a positive score.

**Key Questions For Authors:**

1.	What is the precise conceptual difference between fuzzy alignment and nearby formulations, especially query-based label decoders and prototype matching methods? More specifically, why is the proposed method more than disease-specific evidence aggregation followed by metric scoring?
2.	Why is Gaussian membership the right choice here? Is it a theoretically derived result, or mainly a design choice?
3.	Can the authors clarify the derivation of Lpoint in the appendix? The gradient-stability claim seems inconsistent with Eq. (34), and the stated constant gradient behavior near extreme predictions is not clear from the current derivation.
4.	The ablations suggest that the strongest gains come mainly from the VQ atomic space and the commitment term, while the fuzzy set-level component contributes less. Can the authors better isolate the contribution of fuzzy alignment itself?
5.	Can the authors provide more direct evidence that the quantization step preserves useful information, for example, through codebook usage, quantization error, or sensitivity to compression level?
Overall, I think this is a well-presented work. If the authors can clarify the points above to help me better understand, I would like to raise my score.

**Limitations:**

No. The paper is well motivated, and the overall design is thoughtful. However, the limitations section would benefit from a clearer discussion of two points: whether the main empirical gains are driven primarily by the VQ atomic design rather than the fuzzy set-level component, and whether the quantization step preserves feature fidelity.

**Strengths And Weaknesses:**

Strengths:
1.	The paper addresses an important clinical problem. Multi-label diagnosis with overlapping evidence and disease correlation is clinically relevant and methodologically meaningful.
2.	The method is conceptually coherent. The atomic visual space, semantic anchors, and fuzzy alignment are well motivated and fit together naturally. The paper is clearly written and easy to follow, and the figures are clear and helpful for understanding the overall framework.
3.	The empirical scope is solid. The method is evaluated on three datasets from different modalities and improves the reported metrics across all of them.

Weakness:
1.	The core concept of fuzzy alignment is not yet clearly distinguished from nearby formulations, especially query-based label decoders and prototype matching methods. A direct comparison to a strong query-based label decoder would strengthen the novelty claim.
2.	The choice of Gaussian membership is not fully justified. The paper presents it as theoretically grounded, but the appendix reads more like a motivated kernel choice than a uniquely derived result. It remains unclear why Gaussian membership is the right form here, as opposed to another reasonable distance-to-score mapping. Why choose Gaussian membership?
3.	The appendix analysis of 𝐿point seems problematic. The gradient-stability claim does not appear to follow from Eq. (34). The stated constant-gradient behavior near extreme predictions is not obvious from the given expression, which instead seems to grow as μc approaches 0 or 1.
4.	The ablations suggest that the main performance gains may not come primarily from the fuzzy alignment design itself. Instead, the strongest improvements appear to come from the VQ atomic space and the commitment term in Table 3, while the fuzzy set-level component seems to play a more auxiliary role.
5.	The paper provides ablation and qualitative evidence that the VQ-based atomic space is useful, but it does not directly evaluate whether the quantization step itself preserves feature fidelity. A more complete analysis would include codebook utilization, quantization error, and performance under different compression levels.

---

> ### Author Rebuttal · Authors · 2026-03-31
>
> Thank you for the detailed and constructive feedback. We are encouraged by your positive assessment of the conceptual coherence and empirical scope. Below, we provide point-by-point responses to each comment.
> ***
> **Q1: Fuzzy alignment vs related methods**
>
> **Ans**: Thank you for the insightful question. Query-based decoders use independent classifiers per class, ignoring dependencies. Prototype methods match a global representation to class prototypes via similarity, with implicit competition between classes. In contrast, our FACT perform set-level alignment between evidence and semantic anchors, optimizing a fuzzy objective over non-exclusive multi-label alignment rather than one-to-one matching. This naturally captures visual polysemy and class dependencies, going beyond simple evidence aggregation followed by metric scoring.
>
> Empirically, replacing fuzzy alignment with a query-based or prototype classifier (with VQ fixed) degrades performance, as shown in R3-Table 1:
> > R3-Table 1
>
> |Method|ODIR-5K||||CXR-LT||||
> |:---|:---:|:---:|:---:|:---:|:---:|:---:|:---:|:---:|
> || **mAP**|**F1**|**AUC** |**Avg.**|**mAP**|**F1**|**AUC**|**Avg.**|
> |Query-based|72.36| 90.23 |92.52|85.04|57.21|94.6|92.88|81.56|
> |Prototype-based|53.52|87.46|85.62|75.54|46.38|94.49|90.72|77.20|
> |**ours**|**72.61**|**90.91**|**92.97**|**85.50**|**57.54**|**94.79**|**93.08**|**81.80**|
> ***
>
> **Q2: Justification for Gaussian membership**
>
> **Ans:** Thank you for this question. The Gaussian membership is not heuristic but arises from the smoothness-constrained formulation of fuzzy alignment (detailed in R2-Q2). We note that it is not the only possible choice; however, empirically it consistently achieves the best performance among alternative mappings (R2-Table 1), indicating a favorable balance between smoothness and sensitivity. We will clarify this in the revision.
> ***
>
> **Q3: Gradient behavior of $\mathcal{L}_{point}$**
>
> **Ans:** Thank you for this valuable feedback. Upon re-examination, the previous “constant gradient” statement was imprecise. The gradient has a local amplification term ($\sim \mathcal{O}(1/\mu_c^2)$) and a global normalization term ($\sim \mathcal{O}(1 / \sum r_c)$). When a single error dominates, the normalization scales as $\mathcal{O}(\mu_c)$, yielding an overall gradient $\sim \mathcal{O}(1/\mu_c)$, comparable to BCE. When multiple errors coexist, the normalization increases with $\sum r_c$, down-weighting each gradient. The gradient is thus adaptively normalized, behaving like BCE for isolated errors but more stable when errors accumulate. We will revise the appendix.
> ***
>
> **Q4: Fuzzy alignment vs VQ gains**
>
> **Ans:** Thank you for your valuable suggestion. These two components are complementary. Tab. 3  in manuscript shows commitment loss mainly boosts mAP, while fuzzy set loss improves F1 and AUC. To isolate the effect, we compare VQ + fuzzy vs. VQ + BCE (Tab. 4 #1 and #3 in manuscript). The latter consistently performs better, showing that the gain comes from the alignment formulation, not representation alone.
> ***
>
> **Q5: VQ quantization fidelity**
>
> **Ans:** Thank you for the insightful suggestion. We provide additional evidence that the VQ step preserves useful information.  Across compression rates (8×/16×/32×), performance remains stable on both ODIR and CXR-LT (R3-Table 2), indicating robustness to compression.
>
> > R3-Table 2
>
> |Rate|ODIR (mAP/F1/AUC/Avg)|CXR-LT (mAP/F1/AUC/Avg)|
> |---|---|---|
> |8X|74.09 / 90.62 / 92.94 / 85.88| 57.86 / 94.67 / 93.17 / 81.90|
> |6X|73.42 / 90.09 / 93.06 / 85.52| 58.15 / 94.85 / 93.23 / 82.08|
> |32X (default)|72.61 / 90.91 / 92.97 / 85.50| 57.54 / 94.79 / 93.08 / 81.80|
>
> Codebook analysis shows low quantization error and compact usage (e.g., ~11–12 codes on CXR-LT), suggesting that the learned atoms capture high-level semantics rather than fine-grained details (R3-Table 3).
>
> > R3-Table 3,
> U = total used codes, Uq = average unique codes per image, QE = average quantization error
>
> |Rate|ODIR (U/Uq/QE)|CXR-LT (U/Uq/QE)|
> |---|---|---|
> |8X|9 / 7.38/6.29e-4|12 / 10.38/7.52e-4|
> |16X|7 / 5.25/5.26e-4|12 / 9.13/7.57e-4|
> |32X|7 / 4.63/8.47e-4|11 / 8.09/7.75e-4|
>
> We further increase code utilization via dead-code reset [Esser et al., CVPR 2021]. While this significantly raises code usage and reduces quantization error, the performance gain is marginal (<0.3 Avg.) (R3-Tables 4–5). Overall, these results indicate that the VQ module preserves semantic information and acts as a compact representation, rather than a lossy bottleneck.
>
> > R3-Table 4,
> DCR = dead-code reset
>
> |Method|ODIR (mAP/F1/AUC/Avg)|CXR-LT (mAP/F1/AUC/Avg)|
> |---|---|---|
> |Default+DCR|74.32 / 90.39 / 92.95 / 85.89|57.75 / 94.77 / 93.14 / 81.89|
> |Default|72.61 / 90.91 / 92.97 / 85.50|57.54 / 94.79 / 93.08 / 81.80|
>
> > R3-Table 5
>
> |Method|ODIR (U/Uq/QE)|CXR-LT (U/Uq/QE)|
> |---|---|---|
> |Default+DCR|93 / 30.50 / 2.46e-4|103 / 20.88 / 6.49e-4|
> |Default|7 / 4.63 / 8.47e-4|11 / 8.09 / 7.75e-4|

---

> > ### Author Rebuttal · Reviewer_zL8X · 2026-04-04
> >
> > Thank you for replying, and would like to keep my positive score.

---

> > > ### Author Response · Authors · 2026-04-06
> > >
> > > Dear Reviewer zL8X,
> > >
> > > Thank you for your thoughtful comments and for your positive feedback. We are encouraged that the clarifications and additional analyses have improved the clarity and presentation of our work.
> > >
> > > We have systematically strengthened the key aspects of the work to address the concerns raised. Together, these additions provide a more comprehensive understanding of the formulation and the role of each component.
> > >
> > > Thank you again for your time and effort in reviewing our paper. We would be very happy to address any last-minute questions or concerns you may still have, particularly if there are aspects of the experimental evaluation that you feel could be further improved or clarified. If you find these clarifications helpful, we would greatly appreciate your consideration in your final assessment.
> > >
> > > Best, Authors

---

### Official Review · Reviewer_taeR · 2026-03-10

**Soundness:** 2
**Presentation:** 3
**Significance:** 2
**Originality:** 2
**Overall Recommendation:** 4
**Confidence:** 4

**Summary:**

The authors argue that many existing methods treat the task of multi-label diagnosis as a rigid discriminative classification problem, which may not be appropriate in medical settings because similar visual patterns can correspond to multiple disease labels. To address this, the paper proposes FACT, which reformulates the task as a fuzzy alignment between atomic visual evidence and disease semantic anchors.

**Compliance With Llm Reviewing Policy:**

Affirmed.

**Key Questions For Authors:**

Please see the weaknesses above.

One additional question is in practical settings, training labels may contain noisy or even incorrect correlations, which could affect the model performance. Have the authors considered this issue?

**Limitations:**

The paper does not discuss its limitations. Medical imaging datasets often contain demographic or acquisition biases, which may affect the learned disease correlations. It would be good to discuss it.

**Strengths And Weaknesses:**

# Strengths
  This work points out that in medical images the same visual pattern may correspond to multiple diseases, so treating diagnosis as a rigid classification problem can be limiting. To address it, it is better to find some shared underlying causes of diseases, which is the so-called atomic visual evidence. So the motivation is clear and direct.

However, there are several weaknesses as follows:
# Weaknesses
1. Many parts of the method rely on well-explored methods, such as GCNs for modelling disease correlations and Gaussian distance for fuzzy membership. So the technical contribution is relatively limited.

2. The RKHS derivation seems to reduce to a standard Gaussian kernel in the end, it is therefore not obvious what new theoretical insight this paper provides.

---

> ### Author Rebuttal · Authors · 2026-03-30
>
> We really appreciate the reviewer for the detailed and constructive feedback. We are encouraged by your recognition that the motivation is clear and direct. Below, we provide point-by-point responses to each comment.
> ***
> **Q1: Many parts of the method rely on well-explored methods ...**
>
> **Ans:** Thank you for the feedback. Our contribution lies in reformulating multi-label diagnosis, rather than introducing new components. Existing methods largely treat diseases independently, which limits modeling of visual polysemy and semantic correlations.
> Our FACT instead reformulates the task as fuzzy alignment between visual evidence and disease semantics.
>
> The components (e.g., GCN, Gaussian membership) follow naturally from this formulation. GCN encodes comorbidity into semantic anchors, and Gaussian membership arises from the smoothness-constrained objective.  Importantly, the novelty lies in the objective-level formulation, not in individual modules, which we intentionally keep simple to demonstrate generality.
> ***
> **Q2: The RKHS derivation seems to reduce to a standard Gaussian kernel in the end ...**
>
> **Ans:** Thank you for this important question. Our contribution is not a new kernel, but the principled requirements for fuzzy membership in multi-label diagnosis. We argue that fuzzy alignment should satisfy continuity and consistency, i.e., similar evidence yields similar memberships, with smooth variation in semantic space, enabling polysemy modeling and stable predictions under small perturbations. Under this formulation, the Gaussian kernel arises naturally rather than heuristically. We will revise the paper to clarify this point.
>
> Empirically, Gaussian achieves the best performance compared to several kernels, as shown in R2-Table 1.
> > R2-Table 1
>
> |Method|ODIR| | | |NIH| | | |RSNA| | | | CXR-LT| | | |
> |:---|:---:|:---:|:---:|:---:|:---:|:---:|:---:|:---:|:---:|:---:|:---:|:---:|:---:|:---:|:---:|:---:|
> ||**mAP**|**F1**|**AUC**|**Avg.**|**mAP**|**F1**|**AUC**|**Avg.**|**mAP**|**F1**|**AUC**|**Avg.** |**mAP**| **F1**|**AUC**|**Avg.**|
> |Linear|69.94|89.64|91.82|83.80|51.94| 89.85 | 87.23 | 76.34 | 89.15 | 92.16 | 96.64 | 92.65 | 57.07 | 94.86 | 92.93 | 81.62|
> |Cosine|68.93|89.66| 91.66 | 83.42 | 52.69 | 90.15 | 87.45 | 76.76 | 89.14 | 92.03 | 96.65 | 92.61 | 57.52 | 94.77 | 93.06 | 81.78|
> |Laplacian| 30.98 | 85.36 | 76.20 | 64.18 | 27.63 | 88.70 | 76.04 | 64.12 | 48.66 | 74.80 | 73.15 | 65.54 | 27.64 | 93.88 | 88.11 | 69.88 |
> |Polynomial| 33.85 | 14.64 | 76.47 | 41.65 | 35.76 | 12.58 | 75.17 | 41.17 | 86.97 | 28.46 | 91.22 | 68.88 | 54.57 | 8.60 | 89.99 | 51.05 |
> | **Gaussian (ours)** | **72.61** | **90.91** | **92.97** | **85.50** | **58.95** | **90.44** | **87.81** | **79.07** | **89.72** | **92.97** | **96.90** | **93.20** | **57.54** | **94.79** | **93.08** | **81.80** |
> ***
> **Q3: ... training labels may contain noisy or even incorrect correlations ... affect the model performance ...**
>
> **Ans:** Thank you for your suggestion. Noisy or incorrect correlations may exist in the data.
> The comorbidity graph is only used to refine semantic embeddings via a GCN without directly constraining predictions.
> Final predictions rely on metric-based fuzzy membership computed from visual evidence, limiting the influence of noisy correlations.
> Additionally, representing each disease with multiple visual atoms (instead of a single prototype) and using self-loops helps preserve semantic identity and improves robustness to noisy neighbors.
>
> Empirically, on a noisy CXR-LT sub-task (25k+ training, 26-class clean test), incorporating GCN consistently improves performance (R2-Table 2,3), indicating robustness to noisy correlations. Detailed results are available in R1-Q3 (R1-Table 5–6).
>
> ***
> **Q4: The paper does not discuss its limitations ... often contain demographic or acquisition biases, which may affect the learned disease correlations ...**
>
> **Ans:** Thank you for raising this important concern. We acknowledge that demographic or acquisition biases may exist and could affect learned disease correlations.
> As discussed in Q3, our FACT mitigates this influence in two ways: (1) the comorbidity graph only refines semantic embeddings and does not directly constrain predictions, and (2) the use of multiple visual atoms per disease reduces reliance on any single correlation pattern.
>
> Empirically, we validate this on CXR-LT, a large-scale (370k+ images), 40-class dataset with extremely long-tailed distributions and multi-hospital sources, approximating realistic clinical scenarios (R1-Table 1 and 2). On a noisy training and clean test sub-task, our method consistently improves performance, especially on tail classes (R1-Table 3–6), demonstrating robustness to label noise and potential biases. While direct quantification of demographic or acquisition biases remains challenging, these results support the method’s stability under realistic conditions.
>
> We will explicitly discuss these limitations in the revision.

---

> > ### Author Rebuttal · Reviewer_taeR · 2026-04-01
> >
> > Thanks for the detailed response. I appreciate the clarifications, and they help me better understand your design choices. However, I still feel the level of technical novelty is relatively limited, and I understand this is not something that can be easily addressed within the same work.
> >
> > So, since my current score is already positive, I do not plan to further increase it.

---

> > > ### Author Response · Authors · 2026-04-05
> > >
> > > Dear Reviewer taeR,
> > >
> > > Thank you for your thoughtful comments and for acknowledging that the concerns have been addressed. We sincerely appreciate your careful evaluation.
> > >
> > > This discussion allows us to further clarify the scope of our contribution. Our motivation stems from two key observations in medical imaging: visual polysemy and disease correlations. Building on these observations, we establish a unified framework that reformulates multi-label diagnosis as fuzzy alignment between polysemantic visual evidence and semantic anchors. We further support this formulation through extensive experiments, showing strong overall performance, with additional validation on long-tailed and noisy settings.
> > >
> > > Thank you again for your valuable feedback, which has helped improve the clarity and positioning of our work. We greatly appreciate your careful review and insightful comments!
> > >
> > > Best,
> > > Authors

---

### Official Review · Reviewer_ESVi · 2026-03-11

**Soundness:** 3
**Presentation:** 3
**Significance:** 3
**Originality:** 3
**Overall Recommendation:** 4
**Confidence:** 4

**Summary:**

This paper proposes FACT, a framework that reinterprets multi-label medical image diagnosis as a fuzzy alignment problem, leveraging vector quantization to construct atomic visual evidence and a graph convolutional network to embed comorbidity topology, with a metric-based fuzzy membership function derived from RKHS theory.

**Compliance With Llm Reviewing Policy:**

Affirmed.

**Final Justification:**

Thank you for your detailed answer, which has basically addressed my questions. I will therefore maintain the "weak accept".

**Key Questions For Authors:**

See the weaknesses.

**Limitations:**

Yes.

**Strengths And Weaknesses:**

# Strengths #
1. Framing multi-label diagnosis as a fuzzy alignment problem that explicitly acknowledges visual polysemy and disease correlations is a fresh perspective, moving away from rigid discriminative boundaries.
2. The decomposition into atomic visual space and comorbidity-aware semantic anchors provides a structured way to handle shared evidence and label dependencies, enhancing interpretability.

# Weaknesses #
1. Insufficient analysis of long-tailed/rare disease classes, which is critical in medical applications. Experiments on large-scale benchmarks, such as CXR-LT or PadChest, are recommended.
2. The proposed point-level fuzzy loss aggregates and normalizes error contributions within each group, enabling a single high-error sample to dominate the aggregated group error and suppress the gradients of all other samples within the same group. This mechanism introduces a theoretical risk of amplifying label noise in the context of extremely imbalanced and noisy medical data.
3. The comorbidity graph is constructed directly from dataset statistics, which introduces dataset bias and noise without accounting for the statistical reliability of the estimates. On small datasets or for rare diseases, the conditional probability estimates exhibit high variance, allowing spurious co-occurrences to generate strong associations that are subsequently propagated by the GCN, causing the semantic anchors to be dominated by noise.
4. Analysis of failure cases are not provided.

---

> ### Author Rebuttal · Authors · 2026-03-30
>
> We deeply appreciate Reviewer ESVi for the positive and insightful feedback. We are equally encouraged by your recognition of the "fresh perspective" and "enhancing interpretability". Below, we provide detailed responses to each of the comments.
> ***
> **Q1: Insufficient analysis of long-tailed/rare disease classes ...**
>
> **Ans:** Thank you for this insightful suggestion. While our original datasets are long-tailed (Appendix Fig. 7), they are relatively moderate. We agree that evaluation on more extreme long-tailed data would strengthen the analysis.
> Following your suggestion, we evaluate on CXR-LT (370k+ images, 40 classes) with severe imbalance. We group labels into head (≥10%), middle (1–10%), and tail (<1%) groups (head: 9 / 24.82%, middle: 14 / 2.7%, tail: 17 / 0.4%).
> As shown in R1-Table 1, compared with the top-3 baselines, our FACT is competitive on head classes and shows clear improvements on middle and especially tail classes.
>
> > R1-Table 1 (AUC)
>
> |Method|Head|Middle|Tail|
> |:---|:---:|:---:|:---:|
> |LDR| 77.30 | 74.75 | 75.73 |
> |MultiCo| 76.84 | 70.06 | 64.71 |
> |HydraViT| **78.15** | 76.02 | 77.66 |
> |**FACT (ours)**| 77.88 | **78.41** | **81.20**|
>
> We also report overall performance on CXR-LT (R1-Table 2). Our method achieves the best micro-averaged results (computed over all sample–class pairs), showing that tail improvements do not compromise overall performance.
> > R1-Table 2
>
> |Method|mAP|F1|AUC|Avg.|
> |:---|:---:|:---:|:---:|:---:|
> |LDR|56.68|94.11|92.70|81.16|
> |MultiCo|55.68|94.05|92.20|80.64|
> |HydraViT|57.32|94.58|92.81|81.57|
> |**FACT (ours)**|**57.54**|**94.79**|**93.08**|**81.80**|
> ***
> **Q2: The proposed point-level fuzzy loss ... introduces a theoretical risk of amplifying label noise ...**
>
> **Ans:** Thank you for the valuable feedback.  We agree that emphasizing high-error samples may raise concerns about noise amplification.
>
> With $r_c=\frac{\mu_c}{1-\mu_c}$ or $\frac{1-\mu_c}{\mu_c}$ (Eq. 8), the gradient scales as $\frac{1}{1+\sum r_{c'}} \cdot \frac{1}{\mu_c^2}$ (Eq. 34). The second term amplifies error samples, while the first term normalizes by total error.
> When a single error dominates, $\sum_{c'} r_{c'} \sim \mathcal{O}(1/\mu_c)$, yielding an overall scaling
> $\mathcal{O}(1/\mu_c)$, matching BCE. When multiple errors coexist, the first term increases with $\sum_{c'} r_{c'}$, further suppressing individual gradients, preventing noisy sample from dominating.
> In contrast, BCE lacks such normalization and is more sensitive to isolated noise.
>
> Empirically, on a noisy CXR-LT sub-task (imbalanced training, 25k+ samples; clean test, 26 classes), replacing $\mathcal{L}_{point}$ with BCE reduces overall performance (R1-Table 3), with the largest declines observed on tail categories (R1-Table 4). These results suggest that the proposed loss is robust rather than sensitive to label noise.
> > R1-Table 3
>
> |Method|mAP|F1|AUC|Avg.|
> |:---|:---:|:---:|:---:|:---:|
> |ours w/ BCE|58.28|86.69|83.59|76.19|
> |**ours w/ point**|**58.79**|**87.05**|**84.06**|**76.63**|
>
> > R1-Table 4 (AUC)
>
> |Method|Head|Middle|Tail|
> |:---|:---:|:---:|:---:|
> |ours w/ BCE|74.14|71.13|67.16|
> |**ours w/ point**|73.83|**71.92**|**74.30**|
> ***
> **Q3: The comorbidity graph ... causing the semantic anchors to be dominated by noise.**
>
> **Ans:** Thank you for your valuable question. Data-driven co-occurrence statistics can indeed be noisy, particularly for rare diseases.
>
> Importantly, the comorbidity graph is only used to refine disease embeddings via GCN and does not directly constrain predictions.
> Final predictions are governed by metric-based fuzzy alignment on visual evidence, limiting the impact of noisy edges. Additionally, representing each disease with multiple visual atoms (instead of a single prototype) and using self-loops helps preserve semantic identity and improves robustness to noisy neighbors.
>
> Empirically, we validate this on a noisy CXR-LT sub-task (noisy training, clean test, 26 classes). Results (R1-Table 5 and 6) show consistent gains under noisy supervision. These results indicate that the model remains robust even under noisy correlation estimates.
> > R1-Table 5
>
> |Method|mAP|F1|AUC|Avg.|
> |:---|:---:|:---:|:---:|:---:|
> |ours w/o GCN|57.45|86.36| 83.53|75.78|
> |**ours w/ GCN**|**58.79**|**87.05**|**84.06**|**76.63**|
>
> > R1-Table 6 (AUC)
>
> |Method|Head|Middle|Tail|
> |:---|:---:|:---:|:---:|
> |ours w/o GCN|72.59|68.29|72.87|
> |**ours w/ GCN**|**73.83**|**71.92**|**74.30**|
> ***
> **Q4: Analysis of failure cases are not provided.**
>
> **Ans:** Thank you for your constructive suggestion. We analyze failure cases on CXR-LT and identify the top confusion pairs, which mainly arise from subtle visual differences (e.g., Cardiomegaly vs. Normal, Edema vs. Normal) and contextual ambiguity (e.g., Support Devices vs. Enlarged Cardiomediastinum).
> Notably, “Support Devices” is a non-disease label that introduces visual interference and co-occurs with severe conditions.
> Detailed statistics and visual examples will be included in the revision.

---

> > ### Author Rebuttal · Reviewer_ESVi · 2026-04-03
> >
> > Thank you for your detailed answer, which has basically addressed my questions. I will therefore maintain the "weak accept".

---

> > > ### Author Response · Authors · 2026-04-05
> > >
> > > Dear Reviewer ESVi,
> > >
> > > Thank you for your careful reading and for your positive feedback. We are encouraged that the concerns are now fully resolved.
> > >
> > > The additional experiments conducted in response to your comments, particularly on large-scale long-tailed and noisy datasets, have been incorporated into the revised manuscript to further support our findings.
> > >
> > > Thank you again for your time and effort in reviewing our paper.
> > >
> > > Best, Authors

---

### Decision · Program_Chairs · 2026-04-30

**Decision:**

Accept (regular)

**Comment:**

This paper proposes FACT, a framework that reinterprets multi-label medical image diagnosis as a fuzzy alignment problem, leveraging vector quantization to construct atomic visual evidence and a graph convolutional network to embed comorbidity topology, with a metric-based fuzzy membership function derived from RKHS theory. All the reviewers hold positive comments on the paper. From my view, the paper is in a good quality in both method, writing, etc. I would recommend it as Accepted.